# Context-dependent activation of V1 parvalbumin interneurons enhances visual discrimination

**Lilia Kukovska[1], Katharina A. Wilmes[2], Natsumi Y. Homma[1], Claudia Clopath[3], Jasper Poort [1‡*]**

**1** Department of Physiology, Development and Neuroscience, University of Cambridge, Cambridge, United Kingdom, **2** Department of Physiology, University of Bern, Bern, Switzerland, **3** Department of Bioengineering, Imperial College, London, United Kingdom

‡ Lead contact.
* jp816@cam.ac.uk

## Abstract

Inhibition is critical for balanced cortical activity and learning. Parvalbumin-expressing cells (PV) are the most common cortical inhibitory interneurons. Strong PV activation inactivates cortical regions. However, the effect of moderate activation on vision and dependence on activation strength, timing, and task difficulty is not established. We investigated these three major factors during visual discriminations in mice. Moderate PV activation in the primary visual cortex (V1) improved easy but not difficult discriminations. It did so only during the initial 120 ms after stimulus onset, corresponding to the initial feedforward processing sweep. Both easy and difficult discriminations required undisturbed late phase activity beyond 120 ms, highlighting the importance of sustained V1 activity. Combined optogenetic activation and two-photon imaging showed that behavioral effects were associated with V1 response selectivity changes. A circuit model with nonlinear activation and strong competitive interactions between V1 cells captured the data. This demonstrates that early and sustained V1 activity is crucial for perceptual discrimination and delineates conditions when PV activation shapes neuronal selectivity to improve behavior.

## Introduction

GABAergic inhibitory interneurons shape the activity of excitatory pyramidal (Pyr) cells and are critical for learning, plasticity, and balanced activity in the cortical circuits [1–3]. However, the mechanisms by which cortical inhibition modulates perception and behavior are not well understood.

The primary visual cortex (V1) contains neurons highly selective for visual features such as orientation [4,5], and is required for visual discrimination [6,7]. GABAergic inhibition in V1 can increase feature selectivity [8–12]. It is not yet established which inhibitory cell type is driving these processes. The mouse cortex contains three major types of interneurons [13], expressing parvalbumin (PV), somatostatin (SOM),

**Data availability statement:** The data and code required to generate the results and figures presented in this manuscript are available at Zenodo repository with doi: https://doi.org/10.5281/zenodo.17478714. The simulation code is available at Zenodo repository with doi: https://doi.org/10.5281/zenodo.17396226.

**Funding:** This work was supported by the Wellcome Trust (https://wellcome.org/, J.P., 211258/Z/18/Z), a BBSRC Cambridge DTP targeted studentship (https://bbsrcdtp.lifesci.cam.ac.uk/, L.K., 2279392), and a UKRI Medical Research Council Equipment Grant (https://www.ukri.org/councils/mrc/, MC-PC-MR-X012271/1). The funders had no role in study design, data collection and analysis, decision to publish, or preparation of the manuscript.

**Competing interests:** The authors have declared that no competing interests exist.

**Abbreviations:** CDF, cumulative distribution function; ChR2, Channelrhodopsin-2; FA, false alarm; ITI, inter-trial interval; PFA, paraformaldehyde; PV, parvalbumin; Pyr, pyramidal; ROIs, regions of interest; SI, selectivity index; SOM, somatostatin; SRP, stimulus-specific response potentiation; V1, visual cortex; VIP, vasoactive intestinal peptide.

and vasoactive intestinal peptide (VIP). PV cells are the most common and densely connect to Pyr cells [14,15]. PV cells innervate Pyr cells at the soma and axon initial segment [15,16], and are on average the strongest contributor to inhibition onto Pyr cells [14].

Disrupting the balance of inhibition can be detrimental to sensory perception. Strong inhibition from PV cells silences V1 to abolish behavioral performance [6,7,17,18]. The effect of V1 silencing can depend on the timing of visual processing and task difficulty [19,20]. Delaying silencing by 80 ms after the visually evoked response onset allows discrimination above chance and more difficult discriminations require longer delay [20]. In contrast to strong inhibition, moderate (nonsilencing) inhibition could enhance feature selectivity. A previous study indicated that PV cell activation increased orientation selectivity [21] but other studies found that PV cells influence response gain of Pyr cells without altering orientation selectivity [22–25].

To establish the effect of PV cell activation on excitatory cells and visual discrimination, we combined optogenetics and two-photon imaging, and manipulated three major factors in the same mice—the strength and timing of activation and task difficulty.

We discovered that PV cell activation enhanced performance, but that the effects critically depended on all three factors. Discrimination was only improved when PV cells were moderately activated during the early phase of V1 processing (the initial 120 ms of stimulus presentation) when mice performed an easy task.

In other conditions, despite testing a wide range of stimulation levels, performance was impaired. Delaying optogenetic stimulation relative to stimulus onset by up to 180 ms [20], to target only the late V1 response, reduced the impairment but only partly. These results highlight the importance of sustained undisrupted late phase V1 processing which is also modulated by recurrent and feedback input [26–28]. Interestingly, the manipulations targeting the different processing windows in V1 affected the accumulation of evidence but not the timing of the decision process.

To compare activity in the V1 network during the different conditions, we used simultaneous two-photon calcium imaging from neurons in layer 2/3. Consistent with our behavioral findings, we discovered that moderate PV cell activation in easy discriminations increased selectivity when activation was limited to the first 120 ms and decreased stimulus selectivity when it lasted throughout the entire stimulus condition. In difficult discriminations, changes in selectivity were not correlated to the changes in behavior. Finally, a theoretical circuit model with nonlinear activation and strong competition between V1 cells captured our experimental results.

These findings advance our understanding of the role of V1 and PV cells. Visual discriminations rely on both early and sustained V1 activity. The impact of PV cells on visual discrimination and V1 response selectivity critically relies on the strength and timing of activation and task difficulty, delineating the conditions where PV cell activation shapes neuronal selectivity to improve behavior.

## Results

To investigate the effect of varying the strength and timing of PV cell activation and task difficulty on visual perception, we trained mice to perform a visual go/no-go discrimination task. Head-fixed mice were trained to run on a cylindrical Styrofoam treadmill (Fig 1A) and initiate trials by maintaining running speed above a threshold for 0.5–0.9 s. Each trial consisted of a gray background inter-trial interval (3–6 s), followed by a 2-s presentation of one of two orientations separated by 90° (Fig 1B). Mice were rewarded for licking a spout during the 'go' stimulus. There was no punishment for licking to the 'no-go' stimulus, except during training when stimulus presentation was extended by up to 6 s during a timeout period. Trial outcomes (Fig 1B) were used to calculate behavioral performance (*d'*, see Methods).

Mice initially licked indiscriminately during the gray background inter-trial interval and stimulus presentation (S1A Fig, top panels), resulting in high proportions of hit and false alarm trials (S1C Fig). As training progressed, mice gradually learned to withhold licking to nonrewarded stimuli (S1A Fig, middle panels). Eventually mice learned to exclusively lick in response to the 'go' stimulus (S1A Fig, bottom panels). Running speed also changed stereotypically over the course of training (S1A Fig, right panels) and mice learned to slow down upon stimulus presentation and accelerate for unrewarded

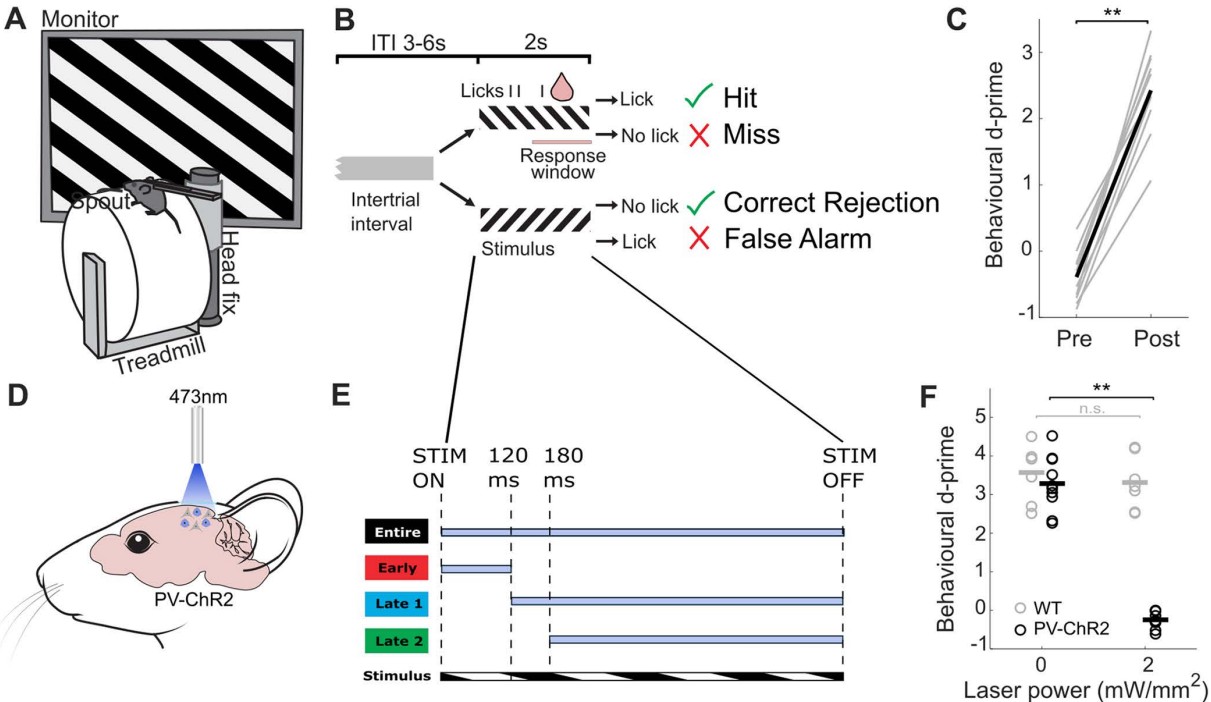

**Fig 1. Learning of a visual discrimination task and optogenetic manipulations in head-fixed mice. A)** Schematic of setup. Mice are head-fixed on a cylindrical Styrofoam treadmill and static gratings are presented on a monitor to the right eye (contralateral to photostimulated side). A spout located in front of the mouse delivers milk during rewarded trials. **B)** Schematic of visual discrimination task. Static grating (2 s) is presented following a variable gray background inter-trial interval (ITI) lasting (3–6 s). For 'go' trials, the reward is delivered when licks are detected during the response window (1–2 s after stimulus onset). Depending on the presence or absence of licks, 'go' trials are classified as hit or miss and 'no-go' trials as false alarms and correct rejections, respectively. **C)** Pre- and post-learning behavioral discrimination performance, *d'*, averaged across mice (Wilcoxon signed-rank test, *p* = 0.002, *N* = 10). Gray lines represent individual mice. **D)** Schematic of optogenetic method. Optic fiber is centered over V1 to deliver blue light (473 nm) to activate Channelrhodopsin-2 (ChR2) expressed in all parvalbumin-positive interneurons of transgenic mice (PV-Cre::Ai32). **E)** Schematic of stimulation protocol. Optogenetic stimulation is delivered relative to stimulus onset (STIM ON) and offset (STIM OFF). 'Entire' condition (black) lasts during the entire stimulus presentation; 'early' condition (red) lasts from stimulus onset to 120 ms; 'late1' condition (blue) lasts from 120 ms to stimulus offset; 'late2' condition (green) lasts from 180 ms to stimulus offset. Each condition was tested in separate sessions (interleaving different power levels). **F)** Discrimination performance, *d'*, in wild-type mice (WT, *N* = 6, 7 sessions) and transgenic mice (PV-ChR2, *N* = 10, 10 sessions) at no laser stimulation (0 mW/mm²) and silencing laser power (2 mW/mm²). Circles, individual sessions; line, mean. See also S1 Fig.

stimuli. Successful discrimination was evident by the significant increase in behavioral *d'* across mice (S1B Fig). Mice performed at chance level in their first training session (−0.39 ± 0.12) and reached an average performance of *d'* > 2 after training (2.43 ± 0.21) (Fig 1C, Wilcoxon signed-rank test, $p = 0.002$, $N = 10$).

**Strong activation of PV cells in V1 impairs discrimination performance**

Strong activation of PV cells is frequently used as a tool to silence cortical activity and study the functional role of brain areas [6,19,29,30]. However, different strengths of PV cell activation in V1 may differentially affect behavior [7,21]. To examine how manipulating PV cell activity over a wide range of laser strengths affects perception, we used photostimulation of PV cells expressing Channelrhodopsin-2 (ChR2) in transgenic mice. We confirmed the average percentage of PV cells that were also positive for ChR2-EYFP was 95.83% ($N = 168$ cells) (S2A and S2B Fig), consistent with previous reports using transgenic mouse lines [14,31,32]. We estimated the boundaries of V1 and sealed all adjacent areas with black paint (see Methods; S3 Fig, example intrinsic imaging map in relation to the mask). We then centered an optic fiber above the craniotomy to deliver blue 473 nm light to V1 (Fig 1D). To exclude the possibility of behavioral effects arising due to nonspecific effects from the light itself [33], we used silencing laser powers in mice lacking ChR2 as a control (Fig 1F). At 2 mW/mm$^2$ laser power illumination, discrimination in control wild-type mice remained high (*d'* 3.57 ± 0.28 to 3.31 ± 0.26, Wilcoxon signed-rank test, $p = 0.297$, $N = 6$, 7 sessions). Moreover, the laser light did not affect pupil size dynamics (S4A Fig). In contrast, discrimination in transgenic PV-ChR2 mice was fully abolished (Fig 1F; *d'* 3.01 ± 0.16 to −0.25 ± 0.07, Wilcoxon signed-rank test, $p = 0.002$, $N = 10$, 10 sessions).

Optogenetically manipulating V1 activity could potentially induce behavioral responses by itself. To determine whether mice responded during PV cell activation in the absence of a visual stimulus, we activated PV cells at different intensity levels in trained mice presented with gray screen. PV cell activation alone lasting for 2 s did not induce licking behavior or changes in running for the duration of stimulation (S4B and S4C Fig).

**PV cell activation during entire stimulus presentation impairs performance at all nonsilencing levels**

Since PV cells are well situated to modify feedforward transmission [34], and have been linked to altering response gain and orientation selectivity, and form stimulus-selective ensembles with Pyr neurons [21,22,25,35], we hypothesized that low levels of activation could sharpen sensory representations to improve discrimination. We therefore tested PV cell activation at a range of nonsilencing laser powers (< 0.5 mW/mm$^2$, see Methods), while mice performed easy discriminations (Fig 2A, 90° angle difference between 'go' and 'no-go' stimuli). When light was delivered throughout the entire stimulus presentation, visual discrimination progressively decreased as a function of laser power. Performance declined from 2.86 ± 0.14 (*d'*, S5A Fig) to between 1.5 and 2 units of *d'* lower at the four highest powers (Fig 2B, 'entire'). As expected [6,7], the strongest PV cell activation at 0.43 mW/mm$^2$ reduced behavior to low performance levels (*d'* 0.91 ± 0.19, S5A Fig), confirming that V1 activity is required for visual orientation discrimination. Interestingly, the effect was mostly mediated by a sharp decrease in licking to the rewarded stimulus (Fig 2C), reaching as low as 0.49 ± 0.08 (hit rate, S5B Fig). Despite a small significant reduction (Fig 2D), licking to the unrewarded stimulus remained relatively stable around 0.22 ± 0.04 (S5C Fig). Notably, individual mice showed variability in false alarm (FA) rate change across laser powers with some increasing and others reducing their licking rate (S5E Fig), suggesting that mice adopted different strategies when presented with the same sensory limitations.

We did not observe an activation level of PV cells that significantly enhanced discrimination. Where previous findings suggested that improvements arise from the significant increase in hit rate [21], in our task, mice already showed a high hit rate (S5B Fig, above 0.95 for 0 mW/mm$^2$). We therefore determined whether improvements in behavioral *d'* were masked by a 'ceiling effect'. We restricted the analysis to include only sessions with a hit rate lower than 0.95 for the baseline (no laser) condition and confirmed our initial findings with no behavioral improvements following PV cell activation (S5D Fig).

 

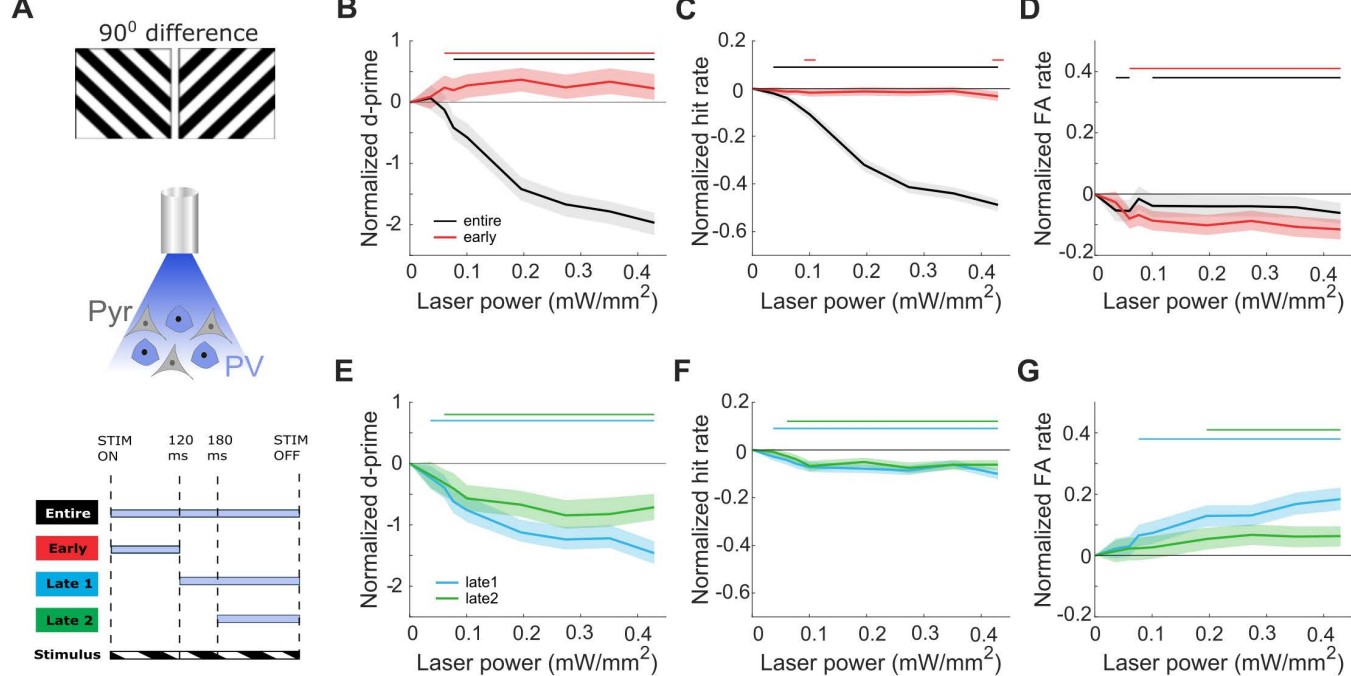

**Fig 2. Stimulating PV cells at nonsilencing levels during entire stimulus presentation or late phase V1 activity impairs discrimination performance, whereas stimulating during early phase V1 activity improves performance.** A) Example of grating stimuli separated by 90° for easy task (top), schematic of optogenetic manipulations activating only PV cells expressing ChR2 (middle), optogenetic stimulation timings for all four conditions aligned to stimulus onset (STIM ON) or offset (STIM OFF) (bottom). **B)** Normalized average discrimination performance, *d'*, **C)** normalized average probability of correct 'go' trials (hit rate), and **D)** normalized average probability of incorrect 'no-go' trials (false alarm rate), as a function of laser power for 'entire' window (black, $N = 10$ mice, 41 sessions) and 'early' window (red, $N = 10$ mice, 37 sessions) conditions. Data is normalized by subtracting the value of the 0 power level (see Methods). **E–G)** same as in B–D) but for 'late1' window (blue, $N = 10$ mice, 40 sessions) and 'late2' window (green, $N = 8$ mice, 32 sessions) conditions. All plots: shaded line is the 95th percentile of data shuffled 1,000 times, using bootstrapping with replacement. Horizontal bars with corresponding colors indicate significant deviation ($p < 0.05$) from no laser stimulation control condition. See also S5 Fig.

## PV cell activation restricted to the early phase V1 activity improves performance at all nonsilencing levels

It has been previously suggested that the earliest visually evoked activity in V1 is sufficient to enable perceptual discriminations [20]. We therefore investigated if moderate PV cell activation restricted to this initial 120 ms following stimulus onset (Fig 2A, 'early'), without interfering with sustained V1 activity that is modulated by recurrent and feedback signals, improved visual discrimination. Indeed, we observed a consistent significant improvement in behavioral *d'* across all non-silencing laser powers (Fig 2B), from $3.11 \pm 0.12$ up to $3.45 \pm 0.16$ (S5A Fig). These behavioral improvements were also present across individual mice (S5F Fig, in 8 of 10 mice), and resulted from a significant decrease in FA rate across all levels (Fig 2D), from $0.20 \pm 0.03$ down to $0.09 \pm 0.02$ (S5C Fig), and a stable hit rate (Fig 2C), with values remaining high in the ranges of $0.95 \pm 0.02$ and above (S5B Fig).

We next wondered whether these perceptual benefits would disappear at high levels of inhibition. Indeed, performance at silencing laser powers was impaired in two out of three mice (S5G Fig), and the third mouse delayed its response for the same duration as the laser stimulation (S5H Fig).

## Undisturbed late phase V1 activity is required for optimal performance

Even at lower hierarchical levels such as V1, visual cortical neurons remain active after participating in the initial feedforward sweep of information following the stimulus onset [27]. While the early response window is thought to encode basic

visual features, activity in the late phase response is more closely associated with the behavioral choice of the animal [27]. Since we found that moderate photostimulation restricted to the early phase V1 activity improved discrimination, we asked whether increasing PV cell-mediated inhibition during the sustained V1 activity window could also benefit performance, by targeting top-down influences or biasing the behavioral choice. To spare the initial feedforward sweep, we restricted light delivery to specific time windows, delayed by 120 ms or 180 ms from stimulus onset, and lasting until stimulus offset (Fig 1E, 'late1' and 'late2' respectively). This approach allowed us to simultaneously investigate the effects of varying the strength and timing of PV cell activation.

Irrespective of the delay with which PV cells were activated, we found that performance dropped gradually with increasing laser power (Fig 2E). Behavioral $d'$ for late1 condition decreased from $3.36 \pm 0.12$ in the absence of stimulation to reach values between 2.0 and 2.4 at the strongest stimulations (S5A Fig). On the other hand, $d'$ for late2 condition decreased less, from similar levels of $3.40 \pm 0.13$ to levels between 2.7 and 2.9 (S5A Fig). This divergence in performance was most evident at laser powers over 0.2 mW/mm$^2$. In both delayed conditions the hit rate was maintained at high levels (although slightly reduced, Fig 2F) over $0.90 \pm 0.03$ across all tested laser powers (S5B Fig), contrasting the entire window condition where hit rate was reduced. The reduced $d'$ was mainly explained by a change in FA rate (Fig 2G). The FA rate in the late1 condition gradually increased over laser powers to double from $0.16 \pm 0.03$ at baseline to $0.33 \pm 0.05$ at 0.43 mW/mm$^2$ (S5C Fig). The FA rate in the late2 condition was maintained at a low rate across all laser powers at around $0.17 \pm 0.04$ (S5C Fig). Interestingly, all mice either sustained or increased their FA rate (in contrast to the entire window condition where mice adopted variable strategies, S5E Fig).

The performance of mice was less impaired when they were given more time to perceive the stimulus without manipulating the activity of V1. A previous study has shown that the first visually evoked spikes in V1 are sufficient for making perceptual decisions, resulting in performance above chance levels [20]. Importantly, our results demonstrate that easy discriminations rely on unaltered late phase V1 activity to achieve maximum performance.

## Difficult discriminations do not benefit from PV cell activation

After establishing the effects of varying the strength and timing of PV cell activation on easy discriminations, we next wanted to determine whether task difficulty influences the effects. While mice can discriminate differences as small as 9° [36], we wanted to have a condition where the discrimination was significantly more difficult but mice were still able to reliably discriminate. Behavioral measurements from 8 mice showed that stimuli separated by 15° were an appropriate choice where performance was lower but still above chance levels (Figs 3A and S6A). When analyzing the distribution of reaction times in the easy and difficult conditions, we observed that the distributions were distinct, consistent with previous reports [37]—difficult discriminations had a broader distribution with longer latencies (S6B Fig). We trained mice on the new stimuli until they achieved $d' > 1.5$ (average performance during testing $d' > 2$, S7A Fig). We found that the effects of the entire (Fig 3B–D) and both delayed (Fig 3E–3G) window conditions were similar to those reported in easy discriminations (Fig 2), with reductions in $d'$ of a comparable size.

After restricting laser stimulation to the early window, we saw no improvements in $d'$ across laser powers (Fig 3B). The hit rate significantly decreased with stronger PV cell activation at powers greater than 0.2 mW/mm$^2$ (Fig 3C), counteracting the beneficial effects of a slightly decreased FA rate (Fig 3D).

Since difficult discriminations require sustained V1 activity [20], we wondered whether longer PV cell stimulation would be beneficial. To test this, we introduced two new windows lasting from stimulus onset to either 180 ms or 340 ms. We found that the changes in $d'$, hit and FA rates were comparable to the standard early window (S8 Fig). Thus, we did not identify manipulations in the strength or timing of PV cell activation that improved performance in difficult discriminations.

To further understand the effects of moderate PV cell activation in V1 on behavior, we investigated the influence on various aspects of the decision-making (Fig 4). First, we investigated whether improvements in discrimination could be explained by a speed-accuracy trade-off strategy [38,39]. Second, we investigated how the laser illumination influenced

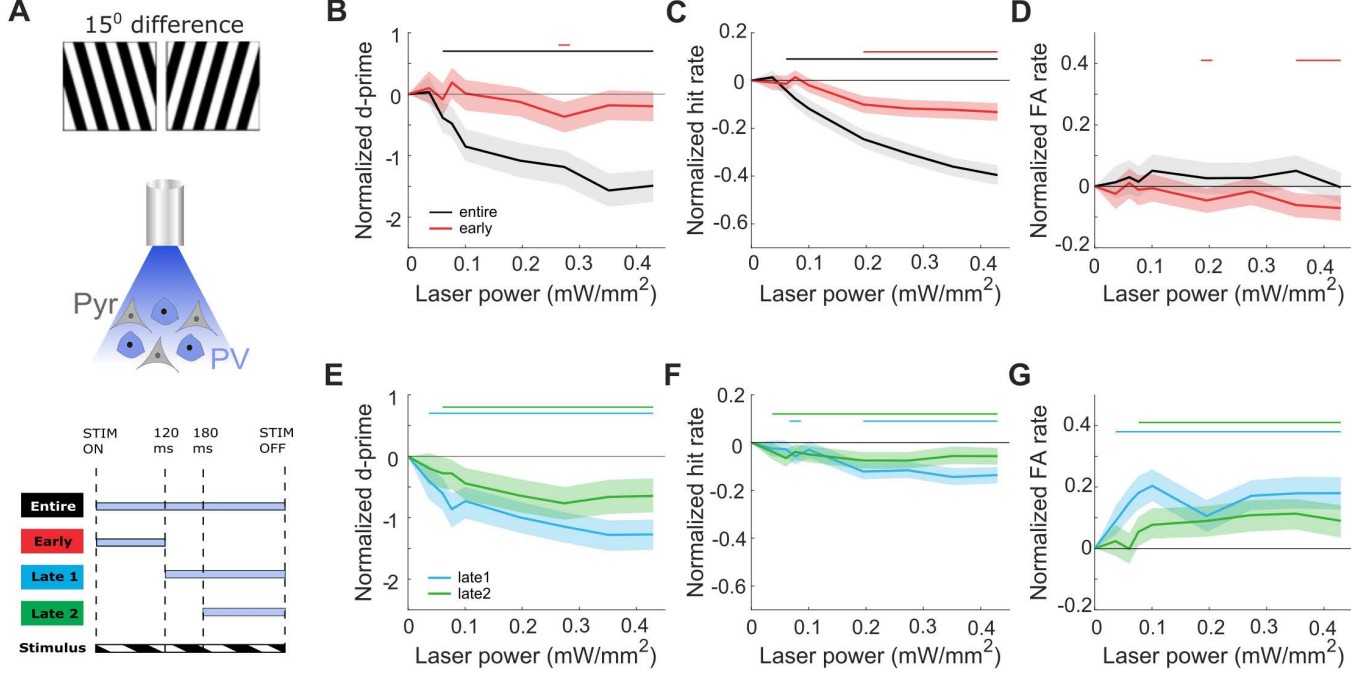

**Fig 3. Stimulating PV cells at nonsilencing levels does not improve performance of difficult visual discriminations. A)** Example of grating stimuli separated by 15° for difficult task (top), schematic of optogenetic manipulations activating only PV cells expressing ChR2 (middle), optogenetic stimulation timings for all four conditions aligned to stimulus onset (STIM ON) or offset (STIM OFF) (bottom). **B)** Normalized average discrimination performance, *d'*, **C)** normalized average rate of correct 'go' trials (hit rate), and **D)** normalized average rate of incorrect 'no-go' trials (false alarm rate), as a function of laser power for 'entire' window (black, *N* = 7, 24 sessions) and 'early' window (red, *N* = 8, 28 sessions) conditions. Data is normalized by subtracting the value of the 0 power level (see Methods). **E–G)** same as in B–D) but for 'late1' window (blue, *N* = 7, 25 sessions) and 'late2' window (green, *N* = 7, 25 sessions) conditions. All plots: shaded line is the 95th percentile of data shuffled 1,000 times, using bootstrapping with replacement. Horizontal bars with corresponding colors indicate significant deviation ($p < 0.05$) from no laser stimulation. See also S7 Fig.

the animal's decision criterion [40]. Third, we investigated the effect of PV cell activation on the profile of licking, running and pupil size. Fourth, we investigated the effect of PV cell activation on different parameters of a simple model describing decision-making dynamics.

## Behavioral improvements are not due to speed-accuracy trade-off strategy

We asked whether the improvements from early window PV cell activation could result from mice adopting a speed-accuracy trade-off strategy, with slower responses reflecting more accumulated evidence and improved performance [38,39]. To quantify reaction times for each trial, we measured lick latencies following stimulus onset (Figs 4A and S9). Although some PV cell activation levels showed changes in the median time to first lick to the rewarded stimulus, delays were minor (20–40 ms, Fig 4A, left panel). Moreover, there were no significant delays in the responses to the unrewarded stimulus (Fig 4A, right panel), despite discrimination improvements arising from reductions in FA rate (Fig 2D). Thus, a speed-accuracy trade-off strategy did not explain the results.

## PV cell activation affects decision criterion

In signal detection theory [40], the sensory process can be characterized by a sensitivity parameter, *d'*, and the decision process by a decision criterion parameter, *c* (see Methods). While *d'* quantifies the separation of two distributions, the decision criterion *c* quantifies whether the animal is more liberal (negative criterion, animal more

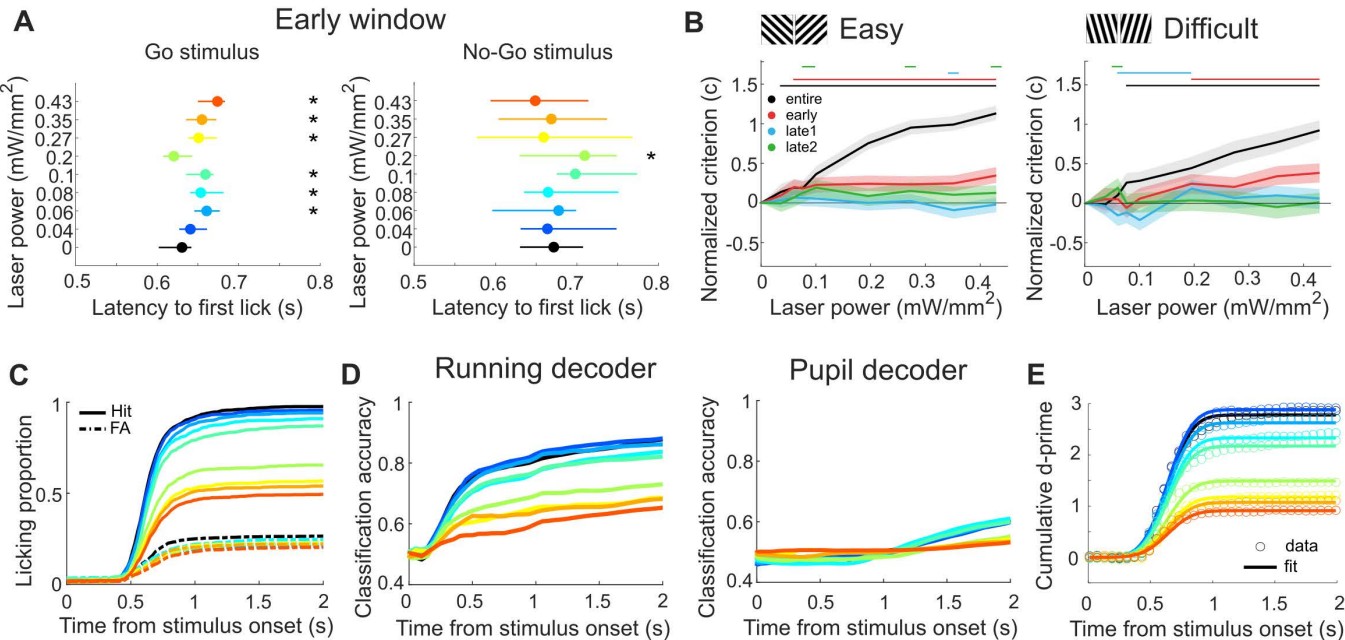

**Fig 4. PV cell activation selectively affects different decision-making components. A)** Response latency for time to first lick in laser off (0 mW/mm²) vs. laser on (0.04–0.43 mW/mm²) in the early window condition (easy discrimination) for 'go' (left panel) and 'no-go' (right panel) stimuli. Circle, median; line, 95% CI of bootstrapping. Asterisks indicate significant deviation ($p < 0.05$) from no laser stimulation control condition. **B)** Normalized decision criterion $c$ across time windows in easy (left panel) and difficult (right panel) discriminations. Shaded line is the 95% CI of data shuffled 1,000 times, using bootstrapping with replacement. Horizontal bars with corresponding colors indicate significant deviation ($p < 0.05$) from no laser stimulation. Data is normalized by subtracting the value of the 0 power level (see Methods). **C)** Licking proportion in response to the presentation of the 'go' (solid line) and 'no-go' (dashed line) stimuli. Plots in C–E show data for the entire window condition (easy discrimination), with colors ranging from blue to red corresponding to increasing laser powers, ranging from 0.04 to 0.43 mW/mm², as indicated in A). **D)** Time course of classification accuracy of running speed (left panel) and pupil size (right panel) linear decoders (probability of correctly identifying 'go' vs. 'no-go' trials), based on the cumulative evidence of running speed or pupil diameter for different PV cell activation levels. **E)** Normal cumulative distribution function (solid line) with fixed mean and standard deviation as a model fit for cumulative $d'$ (circles) across stimulus presentation (for visualization purposes, every second data point is shown). The number of mice and sessions is the same as in Figs 2 and 3.

likely to respond) or more conservative (positive criterion, animal less likely to respond). We investigated whether the changes we observed in $d'$ were associated with changes in the decision criterion. Regardless of task difficulty, the criterion in the entire window condition shifted from negative to positive values, i.e., lick to nonlick trends (from $-0.55 \pm 0.08$ at 0 mW/mm² to $0.61 \pm 0.22$ at 0.43 mW/mm² in the easy task and from $-0.63 \pm 0.10$ at 0 mW/mm² to $0.30 \pm 0.34$ at 0.43 mW/mm² in the difficult task; Fig 4B, black). Smaller shifts in criterion were also observed in the early window condition (from $-0.49 \pm 0.07$ at 0 mW/mm² to $-0.15 \pm 0.08$ at 0.43 mW/mm² in the easy task and from $-0.36 \pm 0.11$ at 0 mW/mm² to $0.10 \pm 0.18$ at 0.43 mW/mm² in the difficult task; Fig 4B, red). We noted that the criterion increased in particular for the entire window condition. We determined whether this was associated with the animal waiting longer with their response. However, a lick latency increase was seen for the go stimulus but not the no-go stimulus (S9A Fig). Furthermore, a significant change in lick latency was also observed in conditions without an increase in decision criterion (S9C and S9D Fig). Thus, with the increased strength of PV cell activation, mice adopted a more conservative strategy in these two conditions. Interestingly, despite interleaving laser powers within recording sessions, criterion values were different, suggesting mice rapidly adapted their behavior on a trial-by-trial basis [41]. Together, these results show that PV cell activation in V1 not only influences perceptual evidence but also the decision processes.

## Similar PV cell activation effects on running speed and licking responses

We next investigated how PV cell activation in V1 influenced other behavioral readouts such as running speed and pupil size. After learning, running shows a stereotypical speed profile, with an early detection component after stimulus onset—a reduction in running speed for both the 'go' and 'no-go' stimulus, and a late component discriminating between the 'go' and 'no-go' stimulus (S1A Fig, right bottom panel). Pupil dynamics have been linked to decision-making performance in other tasks [42,43]. To compare the effects of PV cell activation on running and pupil size with the effects on the licking proportion during both stimulus conditions (Fig 4C) we used a linear decoder (see Methods). The decoder predicted based on the running speed and pupil size the identity of each trial (classifying it as a 'go' or 'no-go' trial), enabling comparison of classification accuracy and licking proportion. We found that the running speed decoder classified trials with accuracy above 80% in the absence of optogenetic PV cell manipulations (Fig 4D, left panel; here shown for the entire window condition). The profile of running speed resembled the profile of the licking proportions (Fig 4C), with a clear separation between laser conditions suggesting similar effects on both behavioral readouts. Interestingly, changes in running speed diverged earlier than the licking proportion responses. In contrast, in our task, pupil size was a poor predictor of stimulus identity, with decoder performance accuracies below 60%, and was not systematically modulated by the laser manipulations (Fig 4D, right panel).

In conclusion, running speed accurately predicts discrimination performance and is modulated by PV cell activation in V1, providing an early readout of the animal's intention to respond to the stimulus by licking. In contrast, pupil dynamics are not a good predictor of behavioral performance.

## PV cell activation modulates amplitude of evidence accumulation

We next determined which function described the dynamics of decision-making and the effects of PV cell activation. We quantified the decision-making dynamics with the cumulative $d'$ (quantifying the $d'$ up to a certain time, see Methods). We found that the normal cumulative distribution function provided good fits with three parameters (the mean describing the onset, the standard deviation quantifying the variability, and the amplitude describing the strength of evidence integration). We created three versions of the model. In model one, the mean, standard deviation, and amplitude were all free to vary across laser powers. In model two, the mean and standard deviation were fixed across laser power conditions. In model three, only the amplitude was fixed. Model two with fixed mean and standard deviation but allowing changes in the amplitude (Fig 4E, here shown for the entire window condition), significantly outperformed the other two models across all time windows and difficulty levels (median $R^2$ of model 2 = 0.995; 2.5th and 97.5th percentiles of model 1 = 0.984 and 0.987; 2.5th and 97.5th percentiles of model 3 = 0.705 and 0.748). In conclusion, the evidence accumulation was well described by a simple function. Interestingly, the onset and variability of evidence accumulation remained relatively consistent across all conditions, while the amount differed across laser powers.

## Changes in neural selectivity reflect behavioral performance in easy but not difficult discriminations

Since improvements in easy discriminations were not due to mice adopting a speed-accuracy trade-off strategy, we wondered whether changes in performance were associated with changes in the selectivity of the V1 network to stimulus. To simultaneously record the activity of V1 neurons while activating PV cells, we repeated the same experiments in a different cohort of PV-Cre mice, where we expressed a calcium reporter GCaMP7s and a red-shifted opsin ChrimsonR in layer 2/3 of V1 (Fig 5A and 5B, see Methods). We confirmed the average percentage of PV cells that were also positive for ChrimsonR-tdTomato was 97.01% ($N$ = 134) (S2C and S2D Fig). We confirmed that red light (639 nm) illumination increased PV cell responses and suppressed Pyr cell responses to the visual stimulus and the effects scaled with laser power (Fig 5C, example cells; S10C Fig, population average). Out of 1,156 Pyr cells, 965 of those cells were significantly modulated by the PV cell activation, accounting for 83.48%. As in the behavioral experiments, we moderately suppressed activity of Pyr cells (S10C Fig; approximately 50% suppression) [21].

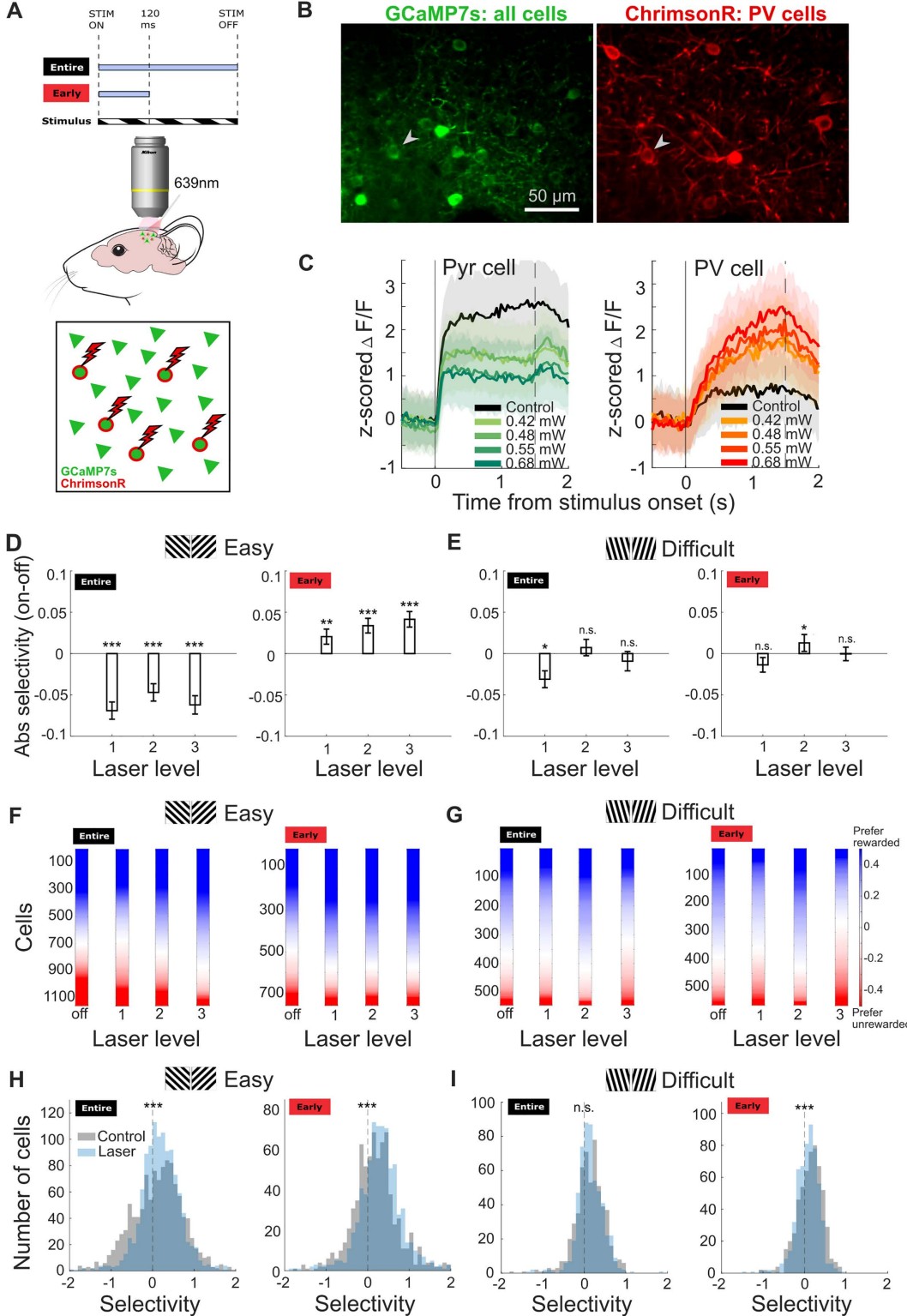

**Fig 5. Changes in neural selectivity caused by PV cell activation reflect behavioral performance in easy but not difficult discriminations.**
**A)** Optogenetic stimulation timings for entire and early window conditions aligned to stimulus onset (STIM ON) or offset (STIM OFF) (top), schematic of optogenetic method combined with 2-photon imaging (middle), schematic of laser activating only ChrimsonR-expressing PV cells while GCaMP7s

reports the activity of all neurons (bottom). **B)** Example frames of in-vivo imaging plane showing GCaMP7s-expressing neurons when illuminated with 920 nm wavelength (left), and ChrimsonR-tdTomato-expressing neurons when illuminated with 1,040 nm wavelength (right). Arrow shows example PV cell. Scale bar, 50 µm. **C)** Average response of two example neurons to optogenetic laser illumination during the entire stimulus duration (0–2 s). The activity to the visual stimulus of the identified Pyr cell is suppressed, whereas the PV cell is increased. Shading, SEM. **D)** Median laser-induced changes in absolute selectivity for entire (left panel, $N = 1,156$ cells, 6 mice) and early (right panel, $N = 754$ cells, 6 mice) time window conditions, for easy task, significant at all laser levels (Wilcoxon signed-rank test, $p < 0.002$). Error bars, SEM. Laser levels in D–G are calibrated for each mouse to cause moderate suppression in Pyr cell activity (see Methods) and range from 0.02 to 0.37 mW. **E)** is same as D) showing entire (left panel, $N = 542$ cells, 3 mice) and early (right panel, $N = 561$ cells, 3 mice) time window conditions but for difficult task (Wilcoxon signed-rank test, 'entire' laser level $1 = -0.03 \pm 0.01$, $p = 0.017$; 'early' laser level $2 = 0.01 \pm 0.01$, $p = 0.037$, all other laser levels n.s.). **F)** Stimulus selectivity index of the same cells as in D) in the no laser condition (0 mW, off) and across laser levels for easy task. Cells were ordered by their selectivity at each laser level. **G)** is same as F) but for difficult task. Color bar range indicates selectivity with positive values for cells preferring the rewarded 'go' stimulus and negative values for cells preferring the unrewarded 'no-go' stimulus. **H)** Histograms of data in F) to quantify the shift towards go or no-go preference in the easy task. Entire: Median $\pm$ SEM, laser off = $0.16 \pm 0.02$, laser on = $0.18 \pm 0.02$, Wilcoxon signed-rank test, $p < 0.002$; averaged across laser levels. **I)** Histograms of data in G) to quantify the shift towards go or no-go preference in the difficult task. Entire: Median $\pm$ SEM, laser off = $0.15 \pm 0.02$, laser on = $0.12 \pm 0.01$, Wilcoxon signed-rank test, $p = 0.931$; averaged across laser levels. Early: Median $\pm$ SEM, laser off = $0.15 \pm 0.01$, laser on = $0.10 \pm 0.01$, Wilcoxon signed-rank test, $p < 0.002$; averaged across laser levels.

Response selectivity in V1 has been previously linked to higher discrimination performance [7]. The selectivity is defined as the difference in the response to 'go' and 'no-go' stimulus divided by the pooled standard deviation, with positive values indicating a preference for the 'go' and negative values a preference for the 'no-go' stimulus (see Methods). We quantified the median of the stimulus selectivity and found that more cells preferred the go stimulus after learning (S10A Fig) [7,35]. We investigated the selectivity changes as a result of PV cell activation for each time window and laser power separately. When mice performed easy discriminations, the absolute selectivity across all laser levels decreased when activating PV cells during the entire stimulus presentation (Fig 5D, left panel; largest change for laser level $1 = -0.07 \pm 0.01$, $p < 0.002$, $n = 1,156$ cells, 6 mice) and increased when restricting activation to the early window only (Fig 5D, right panel; largest change for laser level $3 = 0.04 \pm 0.01$, $p < 0.002$, $n = 754$ cells, 6 mice), reflecting the previously reported behavioral changes (Fig 2B). We confirmed that the observed increases in selectivity were not caused by rebound activity after ending PV cell activation at 120 ms (S10B Fig). This is in agreement with previous reports showing that photoinhibition durations shorter than 500 ms, regardless of laser intensity, causes minimal rebound [44].

Furthermore, the increase in the absolute selectivity in the early window condition resulted from a shift toward the rewarded 'go' orientation (Fig 5F and 5H, Early: median $\pm$ SEM, laser off = $0.18 \pm 0.02$, laser on = $0.31 \pm 0.02$, Wilcoxon signed-rank test, $p < 0.002$, $N = 754$, 6 mice; averaged across laser levels; Note: positive values indicate go stimulus preference, negative values indicate no-go stimulus preference). A match between reductions in behavioral performance and neuronal selectivity was not observed in difficult discriminations. Most laser levels showed no significant changes in absolute selectivity (Fig 5E) and no consistent bias in 'go' or 'no-go' selectivity (Fig 5G and 5I). Thus, our results show that increased behavioral performance during PV cell activation in the early time window during easy discriminations is associated with increased response selectivity in V1.

## Circuit model captures changes in neural selectivity with PV cell activation

How can activating PV cells at different times have such contrasting effects on neural selectivity? And how can brief PV cell activation lead to increased selectivity? To address these questions, we constructed a simplified circuit model that captures the minimal mechanisms required to explain our experimental observations. The model consists of two excitatory populations (Pyr neurons) that compete via lateral inhibition and receive additional inhibition from two PV cell populations, both of which are activated by laser stimulation (Fig 6A). Stimulating the circuit over 100 trials (50 go and 50 no-go) reproduced key features of the data. First, PV cell activation throughout the stimulus period led to a reduction in Pyr cell activity (S11E Fig) and a corresponding decrease in absolute selectivity (Fig 6B). Second, early PV cell activation induced a transient suppression, followed by a gradual rebound in activity (S11F Fig), accompanied by an increase in absolute selectivity (Fig 6B). Third, the

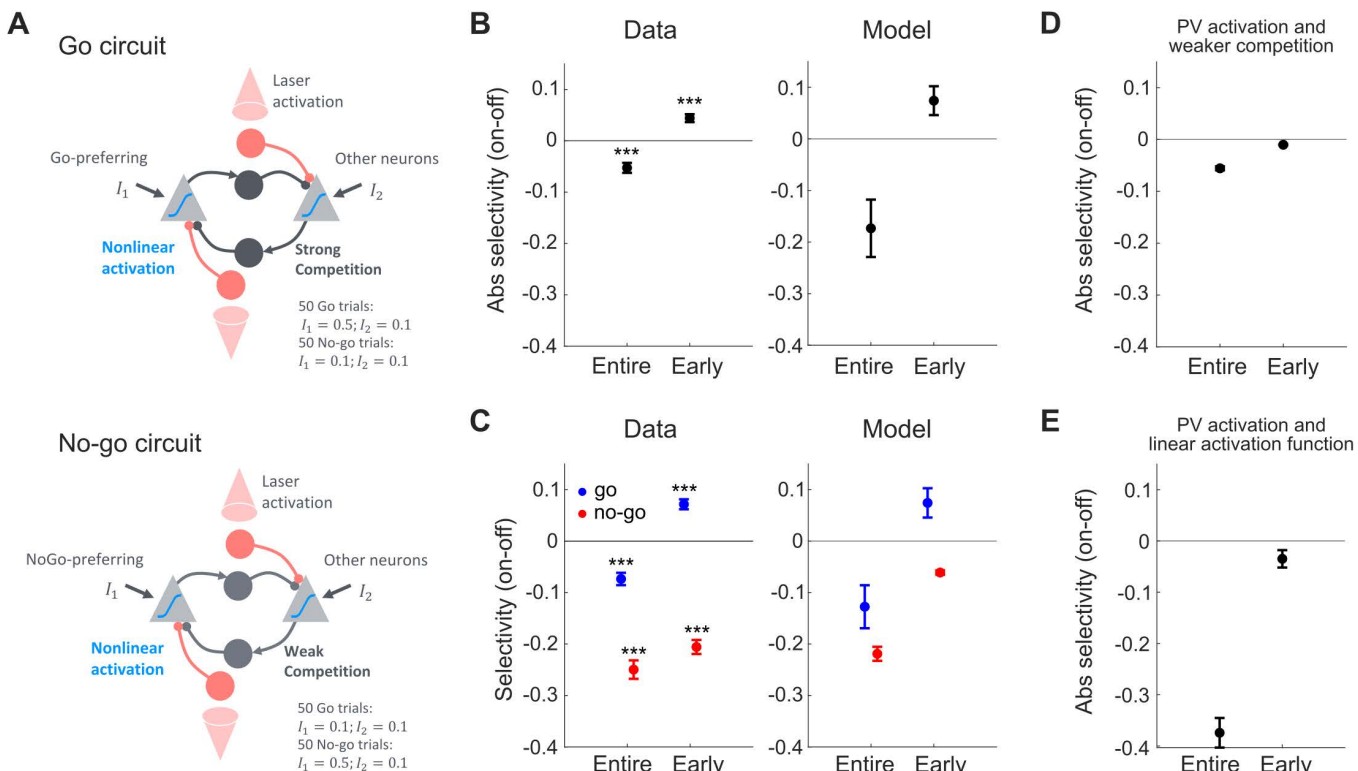

**Fig 6. Increased selectivity requires competition and nonlinear activation function in pyramidal cells. A)** Illustration of the circuit model. Light gray triangles denote excitatory pyramidal populations, circles denote inhibitory populations. Excitatory populations inhibit each other via local inter-neurons (dark gray) and receive additional inhibition from PV cells (red). Top: Go-preferring cells circuit interactions. The left pyramidal population is a go-preferring population, receiving stronger input during 'go' trials. The dark gray connections are strong and hence there is strong lateral inhibition leading to competition in the circuit. Bottom: No-go-preferring cells circuit interactions. The left pyramidal population is a no-go-preferring population, receiving stronger input during 'no-go' trials. The inhibitory interactions between the pyramidal cells are less strong and hence there is less competition in the circuit. **B)** The difference in absolute selectivity between laser on and laser off conditions for 'Entire' and 'Early' PV cell activation for the experimental data (left panel) and the model (right panel). Same data as in Fig 5D but averaged across laser levels. **C)** The difference in selectivity between laser on and laser off conditions for go- (blue) and no-go-preferring (red) populations for 'Entire' and 'Early' PV cell activation for the experimental data (left panel) and the model (right panel). Same data as in Fig 5F but averaged across laser levels. Note that no-go selectivity was multiplied by −1 to indicate increased selectivity if above 0 and decreased selectivity if below 0. **D)** The difference in absolute selectivity between laser on and laser off conditions for 'Entire' and 'Early' PV cell activation in a model with weaker competition. **E)** The difference in absolute selectivity between laser on and laser off conditions for 'Entire' and 'Early' PV cell activation in a model with (rectified) linear activation function. Across all panels: Error bars, SEM.

model captured the experimentally observed asymmetry: the selectivity increase was primarily driven by enhanced selectivity in go-preferring cells (Fig 6C). Two elements of the model proved critical for these effects: competitive interactions among Pyr cells, and a nonlinear activation function in the Pyr cells. In the model, go-preferring Pyr cells are embedded in stronger competitive interactions than no-go-preferring cells, implemented by differences in their inhibitory connectivity. Depending on the strength of this competition, the circuit settles into different steady-state activity patterns, reflected in the more pronounced and persistent change in activity for go-preferring cells following early PV activation. Removing this competition reduced selectivity overall and abolished the selectivity enhancement seen with early PV cell activation (Fig 6D). Likewise, replacing the nonlinear activation function with a rectified linear one eliminated the selectivity increase (Fig 6E), highlighting the importance of nonlinear response properties in shaping the effect.

The model was intentionally kept minimal to allow for mechanistic insight. Although simplified relative to actual cortical circuits, it demonstrates that the experimentally observed effects can be accounted for by basic circuit motifs. The

conclusions were not strongly dependent on precise parameter choices (S11C and S11G Fig). Rather, the essential requirement was that certain qualitative connectivity constraints are met (e.g., strong competitive inhibition). This minimal modeling approach allowed us to isolate the key circuit ingredients responsible for the specificity of PV-mediated modulation.

## Discussion

Cortical inhibition shapes activity of excitatory cells and is critical for learning and plasticity and balanced activity in cortical circuits. However, the impact of parvalbumin-expressing cells (PV)—the most common inhibitory interneuron—on excitatory cells and behavior during visual discrimination has not been established. Our results advance our understanding of the role of the primary visual cortex (V1) and V1 PV cells. First, they demonstrate that both early and sustained V1 activity is crucial for optimal behavioral performance of easy and difficult discriminations. Second, they show that PV cells can enhance visual discriminations and increase selectivity of V1 neurons but that the benefits critically depend on the strength and timing of activation and task difficulty.

Strong activation of PV cells has been used as a tool to silence V1 and study its involvement in behavior [6,17]. Changing the timing of silencing has helped to determine the epochs in which V1 contributes to behavior [19,20]. On the other hand, using nonsilencing levels helps to reveal the function of inhibitory interneurons in shaping the activity of principal neurons and behavior [21,29]. However, the functional role of PV cells has been under debate [21,22,25,45], and how PV cells contribute to the different phases of V1 processing to perturb or benefit visual discrimination is not known. Our study is the first to examine the combined effect of activation strength and timing, and task difficulty with simultaneous optogenetics and two-photon imaging.

Our results show that behavioral improvements were not a result of mice adopting a speed-accuracy trade-off strategy [46,47], which would have influenced both easy and difficult discriminations [38,48]. In our task, mice indicated their decisions by licking a reward spout. We additionally measured the effect of moderate PV cell activation on running speed and pupil size to understand the influence on the decision-making process. Studying movements during behavior provides a continuous readout of the evidence accumulation and the internal decision dynamics [49,50]. We found that changes in running speed preceded licking, accurately predicted the choice of the animal, and were similarly modulated by PV cell activation. Pupil size, previously linked to motivation, decision confidence, and behavioral state [51,52], did not predict stimulus identity. This supports previous work dissociating running from pupil dynamics [52]. The dynamics of these decision processes were well described by a model using a normal cumulative function. Strikingly, the timing and variability of evidence accumulation were preserved despite varying PV cell activity at different times of visual processing. Instead, effects of PV cell activation on decision-making were best described by the modulation of evidence strength.

Traditionally, V1 has been viewed as a region specialized in processing basic visual features such as orientation. However, extensive reciprocal connections between V1 and higher-level visual areas have since been described, suggesting that V1 is a more complex processing area. For example, the activity of V1 neurons can be influenced by behavioral state and running speed [52–54], spatial navigation [55,56], auditory signals [57–59], predictions [60], and task context [61]. Previous work distinguished between an early and late phase of activity, reflecting mainly feedforward and feedback processing [27,28,62,63]. The precise timing of these two phases is not yet established and it may depend on different factors, such as stimulus or task complexity [64]. A recent study in mouse visual cortex suggested that early V1 activity (the first 80 ms) is sufficient for easy discriminations and enables perceptual decisions above chance [20]. However, we found that the late phase V1 activity is still required for optimal performance. During more difficult visual tasks, V1 continues to be critical for longer periods [20], after hierarchically higher areas have been activated [63]. Extending the early window of activation up to 340 ms for difficult discrimination, however, did not improve the performance in our study. The late phase, i.e., the activity beyond the initial feedforward sweep, coincides with the time when recurrent and top-down influences from secondary visual and decision-making areas contribute to the local computations to modulate V1 activity [27,28].

Feedback to V1 improves visual stimulus encoding required for behavior [19]. Additionally, top-down influences enhance their relative impact with learning [65] and carry predictions that can shape neural responses and facilitate perception [66,67]. As a result, activation of PV cells during the recurrent and late V1 activity phase may perturb top-down influences and indirectly impair performance.

Previous studies using a go/no-go discrimination task demonstrated an association between the response selectivity of V1 neurons and behavioral performance [7,68]. We therefore reasoned that the changes in visual discrimination performance following the moderate activation of PV cells could result from changes in the stimulus response selectivity of the local network. We confirmed our hypothesis that improved discrimination in the early window condition was associated with increases in V1 response selectivity. Similarly, the behavioral impairments in the entire window condition were correlated with decreased selectivity. Interestingly, these effects were only visible in easy discriminations, and we did not find a clear relation between V1 selectivity changes and performance of difficult discriminations. Due to their dense connectivity, PV cells may be more broadly tuned than Pyr cells [69–71], which can make their tuning precision insufficient to enhance Pyr cell activity during more difficult discriminations. It is also possible that complex tasks rely more heavily on higher-level areas to influence decisions, for example, on the posterior parietal cortex or the cingulate cortex [72–77]. V1 neurons exhibit strong orientation selectivity [5] and V1 is necessary for orientation discrimination [6,7]. We therefore reduced orientation difference to increase task difficulty similar to previous work [20,21,36]. We also used full field visual stimuli to homogenously drive visual cortical neurons similar to previous studies [7,21,35,68]. Larger or complex stimuli can differentially recruit other inhibitory cell types such as SOM and VIP cells [78,79]. Follow-up studies could compare the effect of PV activation during other types of difficulty manipulations such as contrast, spatial frequency or stimulus size. It would also be interesting to determine how PV activation effects in our task compare for other types of experience-dependent plasticity such as stimulus-specific response potentiation (SRP), which is known to rely on PV interneurons [30], or sequence learning, which has pronounced effects on the later phase of activity [66].

We used widefield optogenetic activation of PV interneurons in V1. Recent work quantifying the profile of such activation showed that inhibition is homogenous and extended across all cortical layers [44]. Our data shows that PV activation improves selectivity and behavioral performance after learning. Furthermore, the behavioral effects were consistent across a range of laser powers. This facilitates future applications to enhance behavioral and neuronal selectivity since it does not require limiting activation to cells with specific response properties or within specific cortical layers. Future studies could determine contributions of specific subpopulations of neurons using two-photon photostimulation [80,81].

We developed a simple theoretical circuit model which indicated that our experimental results relied on competition between Pyr cell populations and a nonlinear activation function in the Pyr cells. Our model demonstrates that our findings can be reproduced in a circuit where go-preferring excitatory neurons are embedded in stronger competitive motifs than no-go-preferring neurons. It is not likely that this asymmetry reflects intrinsic differences in visual encoding, as go and no-go orientations were counter-balanced across animals. Instead, we interpret it as learning selectively strengthening circuit motifs that enhance discriminability of go stimuli. The model predicts this arises from differences in lateral inhibition between pyramidal cell populations, but is agnostic to whether this inhibition is mediated directly by PV cells, or through other inhibitory subtypes such as SOM cells. We chose to model this circuit using a minimal architecture comprising only pyramidal and PV cells, to isolate the essential circuit motifs sufficient to reproduce the experimental effects. This simplicity allowed us to identify which specific circuit properties are necessary to account for the observed increase in stimulus selectivity following transient PV activation. We recognize that the model does not include other inhibitory subtypes (e.g., SOM or VIP interneurons) or detailed laminar structure. This was a deliberate choice, because our current dataset does not include recordings from these populations. Most importantly, the interlaminar and subtype-specific connectivity in mouse V1, as well as the effects of learning, remain incompletely characterized. Creating a more complex model would be insufficiently constrained by data and risk obscuring key mechanisms responsible for the observed effects. Future experimental studies could investigate how different inhibitory subtypes or top-down

inputs interact with the dynamics described here. Future modeling studies could use such data to create a more comprehensive mechanistic model and generate further experimentally testable predictions to reveal how our proposed motif operates within richer network contexts.

Our results highlight the important role that PV cells play in shaping neural selectivity and visually guided behavior. They point to the critical importance of specific time windows of visual processing and calibrated strength of inhibition, and the dependence of behavioral effects on task difficulty. Collectively, these findings underscore the contribution of V1 in both simple and complex visual discriminations and highlight the dynamic nature of cortical inhibition provided by PV cells.

## Methods

### Ethics statement

All experiments were performed at the University of Cambridge in accordance with the Animals (Scientific procedures) Act 1986 and with AWERB (Animal Welfare and Ethical Review Board at the University of Cambridge) and UK Home Office review and approval (PPL numbers PA41D42A4 and PP6681989).

### Animals

Mouse lines were obtained from the Jackson Laboratory (PV-Cre, stock number 008069, and Ai32, stock number 012569), and inbred to maintain the original colony or crossed to create a PV-Cre::Ai32 colony [82]. Sixteen adult mice (11 males and 5 females) underwent surgeries at ages between 7–13 weeks. Animals were co-housed in a 12-hour (08:30–20:30) reversed light/dark cycle room and provided with cage enrichment and standard diet and water ad libitum. During behavioral testing, mice were food-deprived to maintain a minimum of 85% (but typically 90%) of their free-feeding body weight (between 3–5 g of standard food pellets per animal per day).

### Surgical procedure

Mice were anesthetized with 2.5% Isoflurane (1%–2% for maintenance) with 2 L/min oxygen, administered with pre-operative analgesia of 5 mg/kg injectable Metacam, and positioned in a stereotactic apparatus. A subset of mice also received intraperitoneal injections with a 10 mg/kg dose of Dexamethasone the day before and day of surgery to prevent brain swelling. Body temperature was maintained at 36.5–37.0°C throughout the procedure using a heating pad. Ointment (Xailin Night) was applied to the eyes to protect them from drying out. The head was shaved, and a circular piece of scalp was removed to expose the skull which was then cleaned and dried. All mice were fitted with a headpost attached by dental cement (C&B Metabond) to allow head-fixation onto the experimental apparatus. A craniotomy was made over the left hemisphere with the center positioned 2.5 mm left of the midline and the posterior edge aligned with the anterior edge of the transverse sinus. Where viral injections were administered (PV-Cre mice only), Picospritzer (Parker Picospritzer III) was used to inject a total of 800 nL of a 2:18 mixture of AAV-syn-jGCaMP7s-WPRE (Addgene: 104487-AAV1) and AAV-syn-FLEX-rc[ChrimsonR-tdTomato] (Addgene: 62723-AAV5), respectively, in 2 injection sites (coordinates ML2.5 and ML2.3 corresponding to the monocular segment of the primary visual cortex [83]) of 200 nL each at depth of 200 μm and 400 μm from the brain surface. The craniotomy was sealed with a double-layer 4 mm-diameter cranial window (two no. 1 thickness windows glued together with index-matched adhesive Norland #71) to provide stable optical access for photostimulation during behavior [84]. A custom-designed 3D-printed cap was used to protect the cranial window. Mice were monitored for 5 days post-surgery with free access to food and water, with post-operative analgesia of 1.5 mg/ml oral Metacam given in the first 3 days. After full recovery, mice were food-deprived, and habituated to experimenter handling and head-fixation apparatus, after which training) commenced.

## Behavioral setup

Mice were head-fixed onto a Styrofoam cylinder (Fig 1A). Running was measured with an incremental encoder attached to the cylinder shaft (Kübler). Behavioral task trials were initiated when running exceeded a threshold for a specified time (typically 1–2 s). An ITI—gray pre-stimulus illumination, was then presented ranging from 3 to 25 s to discourage timed licking (3–6 s for testing). To avoid excessive licking during training, the ITI was extended if mice licked during the gray screen, and up to 6 s timeout was given when mice licked during the unrewarded 'no-go' stimulus. A reward spout positioned near the snout of the mouse delivered strawberry-flavored soy milk via a pinch valve (NResearch), with licks being detected via a piezo disc sensor. Reward was delivered upon first lick only during a reward zone (1–2 s after the rewarded 'go' stimulus onset) or by an auto-reward trigger at 2 s if mice failed to lick. The stimuli consisted of static black and white gratings with a fixed spatial frequency (0.06 cpd) with 100% contrast and had either 135° or 225° orientation. Stimuli were presented pseudorandomly (no more than four of the same stimulus type in a row). Stimuli were presented with an equal probability of 0.5, except in rare cases where mice failed to show behavioral improvements and the 'no-go' stimulus was biased for brief periods with a 0.7 probability to discourage licking. The rewarded stimulus was counter-balanced across mice. Stimuli were presented on a 27" DELL U2715H monitor positioned at 45° to the mouse body axis and 23 cm away from its right eye. Stimulus presentation, reward delivery, and laser stimulation were controlled by a custom-written Matlab script using the Psychophysics Toolbox [85].

Schematics in Figs 1–3 and 5 are adapted from SciDraw.io, from the following contributing authors: Emmett Thompson (3925987); Ethan Tyler and Lex Kravitz (3926057); Long Li (5496322); and Manish Kumar (4914800).

## Experimental protocol

For 2–3 days, mice were habituated to the cylinder and milk rewards. For 2–3 days, mice learned to run on the cylinder (monitor off) uninterrupted and to lick the spout for rewards without visual stimulation. During discrimination training (1–3 weeks), mice learned to discriminate between 'go' and 'no-go' stimuli. Each training session lasted ~1 to 1.5 hours (usually between 100–400 trials). Mice were tested with optogenetics after achieving $d' > 2$ on 3 consecutive sessions, except for two mice which were tested after 1 such session. This criterion was to prevent overtraining and ensure all four task outcomes (hit, miss, FAs, and CRs) were present for analysis. Overtraining will also result in a 'ceiling effect', possibly masking behavioral improvements resulting from PV cell activation. Three factors were tested: 1) strength and 2) timing of optogenetic stimulation, and 3) difficulty of discrimination. Each testing session had five interleaved laser powers (including 0 mW/mm² to establish baseline $d'$) and lasted until at least 400 trials were completed. Testing of timing was done in a pseudorandom order.

## Optogenetic stimulation

A Stradus VersaLase laser was used to deliver 473 nm light via a 200 μm core, 0.39 NA optic fiber cable. Laser intensity was calibrated using a standard photodiode power sensor (S121C, Thorlabs). All laser powers are reported as mW/mm². A laser power was first identified to fully silence discrimination (Fig 1F). Lower test powers, including 0.04, 0.06, 0.08, 0.10, 0.20, 0.27, 0.35, and 0.43 mW/mm², were then selected to produce detectable changes in discrimination. The size and boundaries of V1 were estimated using intrinsic imaging data from 12 mice corresponding to previously reported boundaries [86]. Black paint was used to seal the window and restrict light delivery to V1. The paint was effective in blocking the laser, with an insignificant leakage of 0.02 μW at the highest testing powers. The optic fiber was centered over V1 at either a 10.7 mm or 17.7 mm distance from the window to deliver uniform illumination. Since the fiber was externally placed, side propagation through the brain to activate neighboring brain areas or to reach the retina to produce nonspecific effects is unlikely [87]. Additionally, to minimize laser light reaching the eyes, the optic fiber was shielded by a custom-designed 3D-printed cone. Light delivery lasted throughout the entire stimulus duration or for specific time windows (Fig 1E).

PLOS Biology

## Behavioral and statistical analysis

Task performance was quantified with behavioral *d'* [40], with higher values reflecting better performance. Behavioral *d'* and decision criterion *c* were calculated by:

$$d' = \Phi^{-1}(H) - \Phi^{-1}(F) \quad and \quad c = \frac{-[\Phi^{-1}(H) + \Phi^{-1}(F)]}{2}$$

where $\Phi$ is the normal inverse cumulative distribution function, $H$ is the rate of hit trials (licking for rewarded trials during the response window only) and $F$ is the rate of false alarm trials (licking for unrewarded trials during entire stimulus presentation). While *d'* captures perceptual sensitivity, the decision criterion *c* is a measure of the subject's internal bias to respond; with negative values indicating high bias for 'yes' responses and positive values indicating high bias for 'no' responses. Lick latencies were determined in trials classified as hit or FA as the first lick detected in the stimulus presentation window (0–2 s). Where the data is normalized, this was done by subtracting the value of the control condition (0 mW laser power) to the laser condition.

No statistical methods were used to predetermine sample sizes, but the number of mice and sessions included in the behavioral experiment is comparable to previous studies [6,7,20,88]. All data are presented as mean ± SEM unless otherwise stated. To compare within-subject PV cell activation effects on *d'*, hit, FA, and criterion to the baseline condition (0 mW/mm²), we used bootstrap with replacement 1,000 times to generate 95% CI [89]. Significant effects were defined as nonoverlap between the no laser condition and the 95% CI for each laser power.

## Eye tracking

Pupil positions were recorded with a Raspberry Pi camera (model 3B+ or 4), where infrared LEDs illuminated the eye. A small number of eye frames (<2%) were randomly selected from the whole recording of each session, and pupil extraction thresholds were manually adjusted. The ellipses to track pupil position and diameter were manually verified by inspecting a random subset of samples from each session. See previous studies [7,90] for more details on eye tracking.

## Decoder analysis

A cumulative decoder was employed to quantify the accuracy with which running speed and pupil size could classify trials as either 'go' or 'no-go' at time *t* relative to stimulus onset [7]. The decoder received inputs of running speed or pupil size aligned to stimulus onset. Thirty trials were taken from each condition to make up the training data, which the decoder then used to construct a model of the running speed or pupil size response to the rewarded and unrewarded stimuli (calculating the mean response $\mu$ across trials for each condition). On the remaining trials, the log-likelihood at time *t* that trial *k* belongs to condition *C* (where *C* is the rewarded 'go', *R* or unrewarded 'no-go' stimulus, *U*) is proportional to:

$$L_{C,k}(t) = -\sum_{Tstart}^{t} (D_k(t) - \mu_C(t))^2$$

where *D* indicates the running speed or pupil size. The trial was assigned as a rewarded stimulus response if $L_R > L_U$, otherwise classified as an unrewarded stimulus response. To obtain the cumulative likelihood $L_c$ at each time point *t*, the summation only included time points from stimulus onset $T_{start}$ to time *t*.

## Modeling cumulative d-prime

The cumulative *d'* was calculated to quantify the *d'* up to a certain time point (by calculating the proportion of hits and false alarms at different timepoints relative to stimulus onset). A normal cumulative distribution function (CDF) was used to fit the experimentally observed cumulative *d'* curves (Fig 4E):

$$CDP(t) = a\left(N(\mu, \sigma)\right)$$

where $N$ is the normal cumulative distribution and the three parameters include a, the amplitude, μ the mean, and σ the standard deviation. Model 1 allowed all three parameters to be varied across laser levels, model 2 only allowed the amplitude to vary (while mean and standard deviation were fixed), and model 3 allowed the mean and standard deviation to be varied (while amplitude was fixed) across laser levels. We used half of the trials for the fitting and the other half of the trials for the testing, using $R^2$ to quantify the quality of the fits. The analysis showed that model 2 performed much better than model 3 and slightly better than model 1, indicating that a model with a mean and standard deviation fixed across laser levels, with an amplitude differing across levels, described the data well. Significance was determined by the overlap between the median $R^2$ of a model and the 95% confidence interval of the bootstrap distribution of the other models.

## Immunohistochemistry and fixed tissue imaging

Immunohistochemistry and fixed tissue imaging were performed similarly to previous studies [35,91]. Mice were anaesthetized by an intraperitoneal injection of Pentobarbital and transcardially perfused with 15 ml PBS with heparin (102 mg/L) followed by 10 ml of 4% paraformaldehyde (PFA) in PIPES buffer (60 mM PIPES, 25 mM HEPES, 5 mM EGTA, and 1 mM MgCl2). Brains were extracted and postfixed in 4% PFA for 24 hours and replaced into PBS/Azide (0.02% NaN3). After embedding in 5% agarose, coronal sections were sliced from the primary visual cortex with 50 μm thickness using a vibratome (VT1000S, Leica Biosystems).

The brain slices were washed with PBS and incubated in 5% normal goat serum (G9023, Sigma) in PBS/Triton X-100/Azide (0.25% Triton X-100, 0.02% NaN3) for 2 hours at room temperature and then incubated overnight at 4°C in primary antibody solution (anti-PV, PV235, Swant, 1:500, in PBS/Triton X-100/Azide). Following primary antibody incubation, slices were washed three times in PBS for 5 min and incubated in secondary antibody solution (goat Alexa Fluor 647 anti-mouse, A21240, Invitrogen, 1:1000, in PBS/Triton X-100/Azide) for 2 hours at room temperature. Slices were then washed in PBS and mounted on glass slides (Menzel-Gläser) with mounting medium with DAPI stain (AB104139, Abcam).

Slices were imaged with a laser scanning confocal microscope (LSM 900, Carl Zeiss) using appropriate excitation and emission filters, a pinhole of 1 AU, and a 20x air objective. A series of z-stack images was obtained at a laser power and gain set to prevent signal saturation. All analysis was performed with Matlab.

## Two-photon calcium imaging

Functional in-vivo calcium imaging was performed using a two-photon scanning microscope with a 3 mm working distance Nikon 16x 0.8 NA objective and Ti:Sapphire laser (Mai Tai HP Deep See, Spectra-Physics, < 70 fs pulse width, 80 MHz repetition rate) at 920 nm, wavelength for calcium indicator GCaMP7s excitation. A 12 kHz resonant scanner (Cambridge Technology) and an FPGA module (PXIe-7965R FlexRIO, National Instruments) imaged at a 30 Hz frame rate to acquire 512 x 512 pixels images covering a 500 x 500 μm field of view. Data was acquired using ScanImage 5.6. A National Instruments DAQ card was used to record triggers for synchronization of neural responses, stimulus presentation, optogenetic stimulation, eye camera videos, licking, and running. Laser power was set to 15–40 mW and was kept consistent throughout consecutive recording sessions. Where identification of tdTomato-tagged PV cells was required, the structural marker tdTomato was excited with a wavelength of 1,040 nm.

Imaging commenced no earlier than three weeks after surgery to ensure stable viral expression. A screening session for each mouse was conducted before data collection commenced. During the initial screening session, ThorCam Software (Thorlabs) was used to identify the injection site coordinates guided by blood vessel identification. Retinotopic and orientation mapping helped identify a responsive site for recording with RFs located near the center of the monitor, confirming that cells in the imaging field where responsive to the center of the screen (corresponding to the monocular

visual field). A brief recording was made for anatomical identification of tdTomato-labeled PV cells, subsequently used for identifying PV cells. Recordings were made from layer 2/3 neurons at a depth of 150–250 μm below the pial surface. Mice were removed from the study if imaging was compromised due to bone regrowth or recording site becoming functionally unresponsive due to overexpression of GCaMP7s, if no single session of $d' > 1.5$ was achieved after extended training, or if neuronal activity was fully silenced even at the lowest possible optogenetic laser power.

## Simultaneous two-photon imaging and optogenetic stimulation

A Stradus VersaLase laser was used to deliver 639 nm light to the imaging site via a 200 μm core, 0.39 NA optic fiber cable coupled with a 20 mm Fiber Optic Cannula. The cannula was positioned at a 30° angle and 7 mm distance from the cranial window and held in place by a micromanipulator. Laser intensity was measured across several levels using a standard photodiode power sensor (S121C, Thorlabs), and calibrated for each animal. Selected test levels ranged between 0.02 and 0.37 mW and caused moderate suppression (~50% decrease in activity) of Pyr cell activity, consistent with previous reports [21]. ChrimsonR-tagged PV cells were stimulated by centering the laser beam over V1 to deliver a uniform illumination. A custom-made 3D-printed cone with attached Blackout Nylon Fabric (Thorlabs BK5) was carefully inserted around the objective and optic cannula to protect the PMTs from external light sources while also eliminating optogenetic laser light from illuminating the eyes.

To allow simultaneous two-photon imaging and optogenetic stimulation within the same neuronal population, the stimulus monitor and optogenetic laser were blanked during the linear phase of the resonant scanner such that the monitor was turned off and the laser was blocked out during the opening of the PMTs shutter, and vice versa.

## Imaging data analysis

Data was pre-processed using Suite2p, where frames were registered to correct for brain motion and active neurons were automatically selected as regions of interest (ROIs) to extract calcium traces for further analysis [92]. After automated detection, all sessions underwent a manual inspection to ensure identified ROIs only included cell somas and excluded axons, synaptic boutons, or other artifacts. Pixels within each ROI were averaged to obtain raw fluorescence time series $F(t)$. The raw fluorescence values for each ROI were then converted to $\Delta F/F$. Neuronal activity was aligned to the onset of stimuli and $\Delta F/F$ was averaged over 0–1 s time window following stimulus onset (excluding the response window 1–2 s where reward is delivered) to estimate the response of a cell to oriented gratings. The Wilcoxon signed-rank test was used to determine if the response to the gratings (0–1 s) significantly decreased or increased relative to the pre-stimulus baseline activity (−0.5–0 s). To calculate the z-scored $\Delta F/F$ in the 0–1 s response window, we subtracted the pre-stimulus response and divided by the pre-stimulus standard deviation. Cells included in Fig 5B and 5C were cells where the z-scored $\Delta F/F$ was higher than 0.5.

## Neural selectivity

A selectivity index (SI) was calculated for each cell as [7,35]:

$$SI = \frac{\overline{R_R} - \overline{R_U}}{S_p^{RU}} \quad where \quad S_p^{RU} = \frac{\sum_{i=1}^{k=2} (n_i - 1)\, s_i^2}{\sum_{i=1}^{k=2} (n_i - 1)}$$

where $R_R$ is the averaged stimulus response (0–1 s) to the rewarded 'go' stimulus and $R_U$ is the averaged stimulus response (0–1 s) to the unrewarded 'no-go' stimulus. The difference is divided by the pooled standard deviation of the responses ($S_p^{RU}$), where $n_i$ is the number of trials in i for $k$ conditions. A positive or negative value indicates cells with a preference for the rewarded or unrewarded stimulus, respectively. To calculate the median selectivity across cells, we

 

computed the absolute of the selectivity of each cell. For each cell, the absolute selectivity during laser manipulations was subtracted from the absolute selectivity during the control condition (no laser).

**Circuit model**

We modeled circuit interactions of go- and no-go-preferring cells, where each circuit consists of two excitatory pyramidal cell populations, which inhibit each other via an intermediate inhibitory population; and two additional inhibitory populations, corresponding to (laser-activated) PV cell populations. The activity of the Pyr cell populations is described by their response $r_i^E$, which evolves over time according to the following equation:

$$\tau \frac{dr_i^E}{dt} = -r_i^E + \varphi(I_i - W_{ij}r_j^I - PV_i),$$

The activity of the inhibitory cell populations, which mediate the lateral inhibition is described by their response $r_i^I$, which evolves over time according to the following equation:

$$\tau^I \frac{dr_i^I}{dt} = -r_i^I + \phi(W^{I,E}r_i^E),$$

where $\tau^E, \tau^I$ are the time constants of the excitatory and inhibitory populations, respectively. $I_i$ is the input to population $i$. Each excitatory population receives a different random constant input $I_i$. Population 1 receives a higher input during trials with its preferred stimulus, but not during trials with the nonpreferred stimulus. For the go circuit in go trials, this input is $I_1 \sim N(0.5,0.1)$ and $I_2 \sim N(0.1,0.1)$. For the go circuit in no-go trials, this input is $I_1 \sim N(0.1,0.1)$ and $I_2 \sim N(0.1,0.1)$. Similarly, for the no-go circuit in go trials this input is $I_1 \sim N(0.1,0.1)$ and $I_2 \sim N(0.1,0.1)$. For the no-go circuit in no-go trials, this input is $I_1 \sim N(0.5,0.1)$ and $I_2 \sim N(0.1,0.1)$. $W_{ij}r_j^I$ is the recurrent inhibitory input from the inhibitory population which is driven by the other Pyr cell population and $W_{ij}$ is the effective synaptic weight from the inhibitory population to the Pyr cell population. In our model, go- and no-go-preferring cells are identical in all intrinsic properties. The only difference lies in their connectivity. Specifically, go-preferring cells are embedded in a circuit with stronger lateral competition compared to no-go-preferring cells. This is reflected by the different weights from the inhibitory to the excitatory population, which were as follows: $W_{ij}^{Go} = 10.0$, $W_{ij}^{NoGo} = 5.0$. $PV_i$ is a PV cell population that targets population $i$ and is activated by the laser. $\varphi(x)$ is a sigmoid activation function:

$$\varphi(x) = \frac{1}{1 + e^{-x}}$$

$\phi(x)$ is a rectified linear activation function:

$$\phi(x) = \begin{cases} 0 & \text{if } x \leq 0 \\ x & \text{if } 0 < x < r_{max} \\ r_{max} & \text{if } x > r_{max} \end{cases}$$

Finally, $W^{I,E}$ is the weight from the excitatory to the inhibitory population. The remaining parameters were chosen as follows:

$$W^{I,E} = 1, \ r_{max} = 10.0, \ \tau^E = 1.0 \text{ and } \tau^I = 0.5.$$

The activity of the PV cell populations depends on the laser condition:

$$PV_i = \begin{cases} 2 & \text{if laser on} \\ 0 & \text{if laser off} \end{cases}$$

The activity of each Pyr cell in each trial is initialized randomly, where we sample from a normal distribution and take the absolute value:

$$x \sim N(0, 1)$$

$$r_i(0) = |x|$$

We simulated three different conditions (1,000 time steps each): (1) no PV cell activation, (2) entire PV cell activation, and (3) early PV cell activation. In the first condition, we simulated the network for 1,000 time steps without intervention. In the second condition, we simulated the laser activation by setting $PV_i = 2$ from time step 400 (after the network reached steady-state) till the end. In the third condition, we simulated the laser activation between time steps 400 and 500. We used a simulation timestep of $dt = 0.05$.

To calculate the selectivity in each circuit in the model, we subtracted the mean activity of the go or no-go population, respectively, in go/no-go trials from the mean activity of that population in no-go/go trials:

$$SI_i^{Go} = \overline{r_i^{Go}} - \overline{r_i^{NoGo}}$$

$$SI_i^{Nogo} = \overline{r_i^{NoGo}} - \overline{r_i^{Go}}$$

In Fig 6B, 6D, and 6E, we calculated the difference in absolute SI between laser on and laser off conditions. In Fig 6C, we calculated the difference in SI between laser on and laser off conditions.

We studied the impact of the competition controlled by the mutual inhibitory connections $-W_{ij}$ by reducing them to 1.0. We studied the impact of the nonlinear activation function by replacing it with a rectified linear one:

$$\varphi(x) = \phi(x)$$

## Supporting information

**S1 Fig. (related to Fig 1) Mice learn a visual discrimination task. A)** Changes in licking and running speed profile over training (early, mid, and late) in an example mouse. Licks (black crosses) are aligned to stimulus onset in rewarded 'go' trials (blue shading, left panels) and unrewarded 'no-go' trials (red shading, middle panels). Red dots, reward delivery triggered by licks during the response window; yellow dots, reward delivery following auto-reward trigger. Average running speed (right panels) aligned to stimulus onset, for rewarded 'go' and unrewarded 'no-go' trials for the same example sessions. Shading, SEM. **B)** Average behavioral performance (*d'*, see Methods) and C) hit and false alarm rates across training sessions. Gray lines, individual mice; error bars, SEM.
(TIFF)

**S2 Fig. Reliable expression of ChR2-EYFP and ChrimsonR-tdTomato in PV cells. A)** Example confocal fluorescence imaging region in primary visual cortex of PV-cre::Ai32 mice showing expression of ChR2-EYFP (left) at cell membrane of PV cells (right). **B)** Percentage of PV cells with expression of ChR2-EYFP at the cell membrane in three mice. The

percentage was similar across mice: mouse 1 = 96.72% (*N* = 61), mouse 2 = 95.15% (*N* = 62), mouse 3 = 95.56% (*N* = 45). **C)** Example confocal fluorescence imaging region in primary visual cortex of PV-cre mouse at injection site, showing expression of ChrimsonR-tdTomato (left) at cell membrane of PV cells (right). **D)** Percentage of PV cells with expression of ChrimsonR-tdTomato at the cell membrane in three mice. The percentage was similar across mice: mouse 1 = 97.44% (*N* = 39), mouse 2 = 98.46% (*N* = 65), mouse 3 = 93.33% (*N* = 30). White arrows indicate identified PV cells. **E)** Confocal microscope image (2.5x objective) of coronal slice showing GCaMP (green) and ChrimsonR (red) overlaid on DAPI stain (blue) showing labeling in V1 across cortical layers 1–6 (with injections at 200 and 400 micrometer). **F)** Example confocal image of horizontal slice showing GCaMP (green) and ChrimsonR (red) fluorescence. Full width at half maximum (FWHM) estimates of spread of fluorescence were similar for GCaMP (median 974 micrometer, 738–1,028, *N* = 5) and ChrimsonR median (763, 733–1,010, *N* = 5). **G)** Example zoomed out in-vivo two-photon microscope image (imaging of neuronal activity was done with increased zoom). Labeled cell bodies were contained within area of ~800 micrometer (*N* = 4 examples).
(TIFF)

**S3 Fig. Example intrinsic imaging map in relation to the mask. A)** Example mask aligned to anterior margin of transverse sinus and stereotaxic coordinates to optogenetically stimulate primary visual cortex (see Fig 1F showing abolished visual performance with high power in all mice). **B)** Example retinotopic map using intrinsic imaging [7,93]. Colourscale indicates −180 (monocular) to 180 (binocular region) degrees spanning 110 visual degrees covered by monitor.
(TIFF)

**S4 Fig. PV cell activation alone does not induce changes in behavioral redouts during the stimulation period (0–2 s). A)** Average pupil size across different laser powers (including 0 mW/mm2) in 'go' (continuous line) and 'no-go' (interrupted line) conditions in WT mice. Shading, SEM. **B)** Lick rate during the presentation (0–2 s) of stimulus only (black) or laser stimulation only (color) in PV-ChR2 mice. Proportion of licks during the baseline period, −1–0 s, was compared to the proportion of licks during the laser stimulation period, 0–2 s, using a Chi-square test and no significant difference was detected in both laser conditions. **C)** Running speed aligned to stimulus onset (black) or laser stimulation onset (color). Based on their licking and running behavior, mice cannot detect the laser stimulation alone during the 2 s period in the absence of visual stimulus. Vertical continuous line, stimulus or laser onset; vertical interrupted line, stimulus or laser offset.
(TIFF)

**S5 Fig. (related to Fig 2) Changes in behavioral readouts as a function of laser power in the easy discrimination task. A)** Average discrimination performance, *d'*, **B)** hit rate, and **C)** false alarm rate for all time window conditions. **D)** Normalized average discrimination performance, *d'*, in the 'entire' window for sessions with lower hit rate (> 0.95). Shading, 95% CI of bootstrap distribution. Data is normalized by subtracting the value of the 0 power level during 0–2 s of stimulus onset. The same normalization method is applied across all normalized data plots regardless of the stimulation window duration. **E)** Normalized false alarm rate in the 'entire' window. Color lines, individual mice (*N* = 10, 41 sessions). **F)** Normalized discrimination performance, *d'*, in the 'early' window. Color lines, individual mice (*N* = 10, 37 sessions). **G)** Average discrimination performance, *d'*, in three mice, where high laser power (2 mW/mm$^2$) was used to silence the 'early' window (6 sessions). As expected, two of three mice showed impaired performance. **H)** Average running speed for 'go' (continuous) and 'no-go' (interrupted) conditions for the mouse which retained high performance when the 'early' window was silenced at 2 mW/mm$^2$. Running was delayed relative to stimulus presentation. In all plots, except D), error bars or shading indicate SEM.
(TIFF)

**S6 Fig. (related to Fig 3) Testing a difficult discrimination task. A)** Behavioral *d'* across the go/no-go discrimination task for the trained (90° angle difference) and test (60°, 30°, 15°, and 10° angle difference) orientations (Wilcoxon

signed-rank test, 15° $p = 0.031$, 10° $p = 0.031$, after Bonferroni correction, N = 8 mice). **B)** Response latency distributions (20 ms bins) across different task difficulty levels, measured by the time to first lick. 25th and 75th percentiles are 550–740 ms for easy (90° angle difference), and 630–900 ms for difficult (15° angle difference) discriminations.
(TIFF)

**S7 Fig. (related to Fig 3) Behavioral parameters in the difficult discrimination task. A)** Average discrimination performance, *d'*, **B)** hit rate, and **C)** false alarm rate as a function of laser power for all time window conditions in the difficult task. Error bars, SEM.
(TIFF)

**S8 Fig. (related to Fig 3) No behavioral improvements in difficult discriminations by extending the early window beyond 120 ms. A)** Normalized average discrimination performance, *d'*, **B)** hit rate, and **C)** false alarm rate as a function of laser power for the original (0–120 ms) and two new (0–180 ms and 0–340 ms) 'early' windows (N = 4 sessions). Data is normalized by subtracting the value of the 0 power level (see Methods). Shading, 95% CI of bootstrap distribution.
(TIFF)

**S9 Fig. (related to Fig 4) Effects of PV cell activation on the lick latency of go and no-go stimuli across the different time windows in the easy discrimination task.** Response latency for time to first lick in laser off (0 mW/mm²) versus laser on (0.04–0.43 mW/mm²). Circle, median; line, 95% CI of bootstrapping. Asterisks indicate significant deviation ($p < 0.05$) from no laser stimulation control condition. The number of mice and sessions is the same as in Fig 2.
(TIFF)

**S10 Fig. (related to Fig 5) Effects of PV cell activation on the activity of Pyr cells across the different time windows and task difficulty conditions. A)** Histogram of stimulus selectivity (positive values: cells prefer rewarded go stimulus; negative values: cells prefer unrewarded no-go stimulus) for pre- and post-learning of the visual discrimination task. Pre: $n = 1,147$, 10 mice, $0.05 \pm 0.02$; Post: $n = 1,467$, 9 mice, $0.14 \pm 0.02$; $p < 0.002$ (Wilcoxon rank-sum test). Median ± SEM. **B)** Average stimulus response activity at different times, ranging from 0 to 120 ms (laser duration), 120–240, 240–360, 360–480, and 480–600 ms, during the early window stimulation (0–120 ms) when stimulating PV cells with highest laser (control subtracted). Error bars, SEM. **C)** Average responses of two population of neurons (left panel, Pyr cells; right panel; PV cells) to optogenetic laser illumination at three different levels during the entire stimulus duration (0–2 s). The activity to the visual stimulus of Pyr cells ($n = 1,156$) is suppressed, whereas PV cells ($n = 39$) is increased. Shading, SEM. **D)** Difference in mean visual stimulus-evoked response with PV cell activation (control subtracted), aligned to visual stimulus onset (continuous line). Cells were ordered by their averaged activity at 0–1 s. Color bar range indicates activity, with positive values (blue) for cells increasing their activity and negative values (red) for cells decreasing their activity as a result of laser stimulation. **E)** is same as D) but for difficult task. **F)** Average activity response to visual stimulus in the absence (black) and presence (red) of laser stimulation, for the entire (left panel) and the early window condition (right panel). **G)** is same as F) but for difficult task. In plots D) to G): dashed line at 2 s for the entire window and at 0.12 s for the early window marks the laser stimulation offset; data shown for the highest laser power.
(TIFF)

**S11 Fig. (related to Fig 6) Phase planes for A)** a circuit with strong competition ($W = 10.0$) and **B)** a circuit with no competition ($W = 1.0$), when the preferred stimulus of population 1 is present and the laser is off ($I_1 = 0.5$; $I_2 = 0.1$; $PV = 0$). The $r_1$ nullcline is shown in blue and the $r_2$ nullcline is shown in orange. For a circuit with strong competition, there are two stable fixed points, where either the rate of population 1 $r_1$ or the rate of population 2 $r_2$ is large. For a circuit with weak competition, there is only one fixed point. **C)** The difference in selectivity between laser on and laser off conditions for go- (blue) and no-go-preferring (red) populations for 'Entire' and 'Early' PV cell activation. Competition in the circuit has been

increased to $W = 13.0$, demonstrating that the conclusions were not strongly dependent on precise parameter choices (see also G). Firing rates over time for 15 simulations of go-preferring populations during 50 go trials with either **D)** no laser stimulation, **E)** entire laser stimulation, or **F)** early laser stimulation. In the no laser condition, the activities end up in one of two fixed points. In the early laser stimulation, most activities end up in the same fixed point. **G)** Phase planes for a circuit with different competition weights varying from 8.0 to 14.0. The $r_1$ nullcline is shown in blue and the $r_2$ nullcline is shown in orange. There are two stable fixed points, where either the rate of population 1 $r_1$ or the rate of population 2 $r_2$ is large.
(TIFF)

## Acknowledgments

We thank members of the Poort and Beltramo labs for valuable discussions, and John McClure Jr for support throughout the study. We thank Riccardo Beltramo for comments on the manuscript, and Elisa Galliano and Edina Horvath-Gulacsi for help with histology and immunostaining.

## Author contributions

**Conceptualization:** Lilia Kukovska, Jasper Poort.

**Formal analysis:** Lilia Kukovska, Jasper Poort.

**Funding acquisition:** Jasper Poort.

**Investigation:** Lilia Kukovska, Jasper Poort.

**Methodology:** Lilia Kukovska, Katharina A. Wilmes, Natsumi Y. Homma, Claudia Clopath, Jasper Poort.

**Resources:** Lilia Kukovska, Katharina A. Wilmes.

**Supervision:** Claudia Clopath, Jasper Poort.

**Writing – original draft:** Lilia Kukovska, Jasper Poort.

**Writing – review & editing:** Lilia Kukovska, Katharina A. Wilmes, Natsumi Y. Homma, Claudia Clopath, Jasper Poort.

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
