## [Editor Report · Decision Letter 0]

7 Mar 2025

Dear Dr Poort,

Thank you for submitting your manuscript entitled "Activation of V1 parvalbumin interneurons enhances visual discrimination dependent on activation strength, timing, and task difficulty" for consideration as a Research Article by PLOS Biology.

Your manuscript has now been evaluated by the PLOS Biology editorial staff, as well as by an academic editor with relevant expertise, and I am writing to let you know that we would like to send your submission out for external peer review.

Once your full submission is complete, your paper will undergo a series of checks in preparation for peer review. After your manuscript has passed the checks it will be sent out for review. To provide the metadata for your submission, please Login to Editorial Manager (https://www.editorialmanager.com/pbiology) within two working days, i.e. by Mar 09 2025 11:59PM.

Kind regards,

Taylor

Taylor Hart, PhD,

Associate Editor

PLOS Biology

thart@plos.org

---

## [Decision Letter · Decision Letter 1]

2 May 2025

Dear Dr Poort,

Thank you for your patience while your manuscript "Activation of V1 parvalbumin interneurons enhances visual discrimination dependent on activation strength, timing, and task difficulty" was peer-reviewed at PLOS Biology. It has now been evaluated by the PLOS Biology editors, an Academic Editor with relevant expertise, and by several independent reviewers.

In light of the reviews, which you will find at the end of this email, we would like to invite you to revise the work to thoroughly address the reviewers' reports.

The reviewers say that this study addresses an important question and will be of wide interest. However, the reviewers had concerns about how little raw data is presented, as well as a number of missing controls and analyses, and the lack of biological relevance for the modeling component. Reviewer 3 also pointed out that it was not clear how relevant these circuit manipulations would be for animals in a more naturalistic context. Based on these reports, we think that your study could be raised to a publishable level for PLOS Biology after an extensive revision which thoroughly responds to the reviewers' concerns, including new analyses and textual changes.

After discussion with the Academic Editor, we think that any revised manuscript should include the following changes in particular:

1. Provide more raw data in addition to better contextualize the processed data that you have already included

2. Perform additional controls to address concerns related to inter-individual variability in viral infection/transgene expression

3. Discuss the relevance of your findings for more naturalistic contexts

4. Include a new model that is more biologically relevant, taking account of known circuits and connectivity patterns, or remove the modeling component

Given the extent of revision needed, we cannot make a decision about publication until we have seen the revised manuscript and your response to the reviewers' comments. Your revised manuscript is likely to be sent for further evaluation by all or a subset of the reviewers.

**IMPORTANT - SUBMITTING YOUR REVISION**

*Re-submission Checklist*

*Published Peer Review*

*PLOS Data Policy*

*Blot and Gel Data Policy*

Sincerely,

Taylor

Taylor Hart, PhD,

Associate Editor

PLOS Biology

thart@plos.org

REVIEWS:

Reviewer #1: The role of PV cells shaping visual perception is murky. This is an important area of investigation. The authors attempt to tease out the relation between PV cell activity and visual perception via a behavioral go/no-go licking task performed while optogenetically activating PV cells with different intensities of laser stimulation, use 2P imaging to assess cell selectivity under different conditions, and propose a model to explain their findings. While the authors do show that behavioral measures are differentially impacted by optogenetic activation of PV cells occurring in an early window during visual stimulation compared to later activations, and depend on the difficulty of the task, the 2P data is much weaker as presented (due largely to the decision to show only highly-processed results) and the model is sufficiently abstract/unmotivated that it is hard to credit it as capturing the underlying biology in a meaningful way.

Specific responses follow in roughly chronological order:

* What percent of PV cells are infected by ChR2? Is this consistent across animals and, if not, how are you controlling for this in your analysis and data interpretations? There is no histology or reporting of these types of numbers which can be important for assessing experimental methodology and comparison with other papers.

* The "early" optogenetic stimulus case could cause rebound spiking in the late period that would interfere with task performance. This general effect has been reported in different circumstances and can have a real impact on network activity, even in some cases initiating epileptic activity via rebound spiking in pyramidal cells (Sessolo et al. J Neuro 2015). The authors should determine the extent to which this is happening in their data and whether it could account for some of their findings that differentiate the early and late stimulation cases

Fig 1S E is interpreted to show that mice cannot detect laser stimulation alone based on an absence of licking within the first 2 seconds. There is, however, a clear increase in licking after 2 seconds for the higher laser power (0.9 mW/mm2). This is hard for me to interpret because it isn't reported which laser stimulus protocol is being used in this figure (I assume it's either entire or one of the lates) but assuming that the laser is on for 2 sec, it does seem to show that there is a behavioral consequence of rebound activity when the PV block is released. It would be surprising if mice licked to a sudden absence of visual perception as PV cells activated when they had been trained to lick only when presented with a specific target orientation. The fact that the mice do start to lick after the laser turns off does not accord well with the broader conclusions from this panel and is potentially interesting if it indicates a "rebound perception" of the target grating- in essence activating what you describe as a "stimulus-selective ensembles with Pyr neurons" on page 5 in a manner sufficient to drive behavior. I'm also not sure how to reconcile this with the conclusion that mice adopt a more cautious strategy in the presence of stimulation.

* Fig S2 and it's caption need some work for clarity. For example, what does "normalized average false alarm rate in the 'entire' window" mean compared to the 'early' window mean? This wording seems to suggest that the false alarm rate was calculated based on licking activity during the period of laser stimulation which is not what I think is actually meant based on the methods. It would also help to put title labels on these plots so that I don't have to keep going back and forth to the caption to know what is shown in each panel (same true in other figures too).

* Fig 3, I agree that your data provides performance-based evidence that mice are better at discriminating 90° than 15°, but there are reports that mice can reliably discriminate at 9° difference (Lyamzin et al PNAS 2021) or even lower. More discussion/analysis of the extent to which this task is actually hard as opposed to simply harder, and what is known about PV cells' contributions to this, would be appropriate and help interpret overall conclusions.

* Fig 4, is there a latency difference between the late1 and late2 cases? The authors use latency to show that the mice don't seem to be using a strategy of waiting to accumulate more evidence in the "early" case, but it's not clear what the distribution of lick times looks like for the late1 and late2 cases. Given that there is only a 60 ms difference between the two cases, it's surprising that they separate as much as they do and I'd like to see if there is any delay to lick created by the late stim cases.

Fig 4A - x label should be more descriptive, maybe "latency to first lick"

Fig 4B, I don't understand the stats - this is not normalized as done previously so I can't use 95% CI around zero to see how the data differs from the null case directly. Normalized to match your previous procedures or a plot of the non-laser data would help with transparency on this stat.

Fig 4B, it's notable that the "entire" case is so different from the other cases. Worthy of a comment and some speculation as to why this is so. Are they waiting later

* There is so little 2P data shown that it's hard to know what to make of it. Figure 5 shows only average z-scored data from only 2 example cells out of 754 that went into the data set. There are no supplemental figures showing what raw data looked like in any way leaving me with no ability to independently assess the GCAMP expression levels, how much of the population was responding to stimulation, the extent to which the dynamic responses to stimulation in these 2 cells are representative of the broader population, the percent of cell that were selective and the heterogeneity of PV modulation across this population, where imaging was occurring relative to V1m/b boundaries, etc. Even ignoring this and accepting the data & conclusions as presented, I'm still confused by these results since only the easy task shown a significant effect of PV activation on selectivity even though stimulation had an effect on behavior measures show previously. I would like to see how cellular activation dynamics (as in panel C) differ when the "early" laser activation is used and how they differ for easy and hard stimulus classes. This figure is asking me to take way too much on faith and presenting the data in such a processed/sanitized form that I can't assess its validity or gain any insight into neural activity beyond the one specific metric the authors focus on.

Why do so many more cells prefer the rewarded stimulus? Is this something intrinsic about the orientation selected, a consequence of training, or something more dynamic during the testing?

* Likewise with Figure 6 and the model, which creates a few discrete populations of neurons preferring go or no-go stimuli and introduces competition via "strong" mutual inhibition. Activation of a second class of inhibitory "PV" cells which synapse only on excitatory cells and drive down their activity. The model, as constituted, does capture broadly some aspects of the neural data from figure 5. However, the model is not particularly well motivated or explained in the text of the manuscript making it difficult for me to accept that it is a reasonable model of what is going on in V1. Some specific questions I had that the manuscript doesn't answer: Are the Go/No-Go preferring cells in the same model are different? If different, why? Am I to interpret the Go-preferring input as reflecting orientation selectivity? Why is competition weaker between "other neurons" and No-Go preferring cells than G cells? Why do the PV cells only target Pry neurons and not other inhibitory cells as in V1? How were any of the parameters chosen and is the model sensitive to the specific parameterization? This model is very simplified compared to the actual V1 circuits, I need more explanation of how/why the simplifying assumptions were made and how dependent the conclusions are on these decisions.

* There should be some consideration/discussion of how plasticity, specifically SRP (e.g. Cooke and Bear J. Neuro 2010), might impact task performance. The training period is almost certainly driving SRP in your animals and changing the perception of familiar patterns. This plasticity is known to effect behavior in an orientation-dependent manner (see the "vidget") with a specific requirement for PV cells (Kaplan et al Elife 2016). Particularly relevant to your work, the consequences of plasticity on single cell firing are especially evident in the late sustained response period in SRP and also in a methodologically similar sequence learning task (Price et al Cereb Cortex 2024).

Minor:

Fig S1A - the three vertically arranged rows seem to indicated changes with training but this isn't specified in the caption or labeled

Fig S1E - what is the running behavior here? Following my comments above about rebound activity, I'd like to see if the mice modulate their running after 2 seconds for the high laser power level.

Reviewer #2: The manuscript by Kukovska et al. addresses the question of how inhibition shapes the activity of V1 as a function of the timing and intensity of inhibition itself. The manuscript addresses an important question in systems neuroscience and is generally very carefully done. I only have a couple of comments that I think the authors need to address.

MAJOR

1) I have some trouble understanding the logic of the circuit model introduced in Fig. 6. The authors in fact do not justify how the model architecture was selected, or how it related to the actual cortical microcircuitry, in which multiple neuronal subtypes (and not only Pyr and PV+ neurons) form a complex network that actually implements the experimentally observed results. Given the apparently arbitrary nature of the model, I recommend that the authors either remove this part of the manuscript or replace it with a biologically plausible model incorporating different cell types, layers, etc. Given that a few of such models are available open source, this appears to be feasible, but I leave it up to the authors to decided whether to add a major modeling component (likely more suitable for a different manuscript) or instead expand the Discussion section with a section on the possible circuit-level mechanisms enabling the realization of the observed effects.

2) Related to the previous point, the Discussion section does not address the issue of how the observed experimental effects might be mediated by microcircuit-level mechanisms (e.g. other interneuronal subtypes, layers, etc.). This is an important aspect that should be addressed.

MINOR

1) The authors chose 120 and 180 ms post stimulus onset as time points for their optogenetic inactivation. While I think it's an elegant and parsimonious approach, the question remains about the precise time point at which the effect of inactivation will switch from an "early" to a "late" motif. However, I believe that the authors' definition on the early window might be too simplistic. In fact, several studies indicated how the requirement of V1 for visual processing varies as factors such as the complexity of visual stimuli, but also task complexity itself. The authors base their choice of window only on a couple of papers, but a discussion of whether the duration of the early window might be fixed or dependent of other factors (i.e. determined purely by fixed circuit-level architecture or instead dynamically adjusted by ongoing cortical processing) should be discussed, if only to point out the fact that the actual duration of the "early" window might co-vary with factors such as task difficulty (something that was not explored in the present study).

2) The plots presented in Fig. 5F-G are unclear. While a shift in preference of neurons can be seen, the chosen display makes it difficult to appreciate the magnitude and significant of this shift. A method enabling a quantification of this shift would be preferable.

Reviewer #3: Kukovska et al., investigate how parvalbumin-expressing inhibitory neurons (PV) influence visual discrimination in mice. The research primarily examines how varying both the activation strength and timing of PV neurons affects visual orientation discrimination across two difficulty levels.

The manuscript is clearly written and the conclusions will be of wide interest to the community. Overall, I believe that most experiments were well-conceived. However, several areas would benefit from additional controls and/or expanded discussion to strengthen the interpretation of results:

Major concerns:

1)

The extent of optogenetic driving of PV cells under different laser powers is not established or self-evident from the data presented. For example, while the behavioral effects seems to follow increasing laser power, the example PV cell shown in Fig 5C seems to show no meaningful difference in Z-scored deltaF across different powers. The change in PV cell activity and the corresponding change in Pyr cell activity across laser intensities should be quantified, and ideally verified with electrophysiology to have a meaningful metric of spike output from PV cells under optogenetic drive. Additionally, since PV cells are distributed throughout all layers, varying the laser power can selectively activate PV cells at different depths. These aspects need to be tested and clarified to have solid conclusions about the role of PV cell on visual discrimination.

2)

The conditions of the experiments make extrapolating to the "normal" function of the cortex, i.e. visual circuits under more naturalistic sensory inputs. While this is hardly a novel criticism and a fault common to all our experiments, it should be addressed in the discussion. For example the experiments use full field visual stimulus which reduces drive on both Pyr cells and PV cells from SST neurons, a significant confounder in estimating the output of PV cells in these experiments (and also why concern 1 is so critical). Additionally optogenetic driving of PV cells will have distinct population effects to how PV cells operate in the visual cortex, with almost all PV cells being equally driven at nearly simultaneously. This may very well negate established cortical circuits for sharpening feature selectivity such as lateral inhibition in a winner-takes-all circuit. Along a similar vein, the choice to explore difficulty along the dimension of angle discrimination rather than say contrast also affects to which visual tasks the result are applicable to. These issues should be acknowledged in the discussion.

3)

It is difficult to compare the effects across different cohorts of mice that use different genetic strategies for optogentic expression. For the 2P experiments to be relevant to previous PV-Cre ChR2 experiments the author must establish the spread and penetration of their viral strategy. The examples shown in Fig 5B clearly show ChrimsonR labelled PV without any detectable GCaMP expression. How many of all the PV cells expressed ChrimsonR and GCaMP? How far was the viral spread in V1?

Minor concerns

a)

In Figure 5 simple deltaF traces of individual responses which include the end of the visual stimulus would improve the figure and help reassure the reader.

b)

In the introduction the authors write " By innervating Pyr cells at the soma and axon initial segment, PV cells are strategically positioned to facilitate neuronal processing ",

this is nonsense as all inhibitory synapse (and really all synapses) are in a position to facilitate neuronal processing.

c)

Figure 5F was difficult to read, the color bar scale needs to be much larger and easier to read

d)

Examples of the V1 / HVA maps from the Intrinsic imaging should be included in the supplemental figures.

e)

The text in most figures is extremely small and difficult to read. Font size should be increased and standardized throughout the manuscript.

---

## [Decision Letter · Decision Letter 2]

9 Oct 2025

Dear Dr Poort,

Thank you for your patience while we considered your revised manuscript "Activation of V1 parvalbumin interneurons enhances visual discrimination dependent on activation strength, timing, and task difficulty" for publication as a Research Article at PLOS Biology. This revised version of your manuscript has been evaluated by the PLOS Biology editors, the Academic Editor, and the original reviewers. We would like to apologize again for the delay in sending this decision.

Based on the reviews, we are likely to accept this manuscript for publication, provided you satisfactorily address the remaining points raised by the reviewers. Please also make sure to address the following data and other policy-related requests.

IMPORTANT: Please ensure that you address the following editorial requirements:

----------------

**Title:

We would like to choose a less granular title for your paper. Is this modified version acceptable to you?

"Context-dependent activation of V1 parvalbumin interneurons enhances visual discrimination"

**Financial disclosure statement:

-- Please add links to the funding agencies in the Financial Disclosure statement in the manuscript details.

**Ethics:

-- Please include the full name of the IACUC/ethics committee that reviewed and approved the animal care and use protocol/permit/project license. Please also include an approval number.

**Data:

-- You wrote that data and code required to generate the results and figures will be uploaded and made available upon publication. Please make these items available now and provide a link so that we can examine them.

-- Please ensure that your data availability statements in the form and paper are consistent and accurately reflect where the data can be accessed.

-- As part of this, please supply the numerical values either in a supplementary excel file or as a permanent DOI’d deposition for the following figures:

1CF

5DE

6BCDE

S4G

S10B

S11C

-- Please ensure that all data files are uploaded as 'Supporting Information' and are invariably referred to (in the manuscript, figure legends, and the Description field when uploading your files) using the following format verbatim: S1 Data, S2 Data, etc. Multiple panels of a single or even several figures can be included as multiple sheets in one excel file that is saved using exactly the following convention: S1_Data.xlsx (using an underscore).

**Code availability:

-- Thank you for providing the underlying code in GitHub. However, because Github depositions can be readily changed or deleted, please make a permanent DOI’d copy (e.g. in Zenodo) and provide this URL in the manuscript and Data Availability Statement.

----------------

We expect to receive your revised manuscript within two weeks.

*Published Peer Review History*

*Press*

Sincerely,

Taylor

Taylor Hart, PhD,

Associate Editor

thart@plos.org

PLOS Biology

Reviewer remarks:

Reviewer's Responses to Questions

Reviewer #1: Overall, the authors have been diligent in addressing and responding to my questions and critiques. I particularly appreciate the inclusion of new supplemental figures, particularly S10, which provide significantly more insight into the methodology and underlying neural activity. Likewise, the extended discussion better situates the work in the broad context of other studies and I have no major concerns with the data as presented.

Minor issues:

The highly simplified model of cortical circuits now explicitly represents a deliberate choice to demonstrate that effects similar to those seen in the data can be achieved by a particular circuit motif which is relatively insensitive to parameterization (fig S11). While the motivations for the model are now more obvious, and the simplifying decisions fit generally within computational norms, I remain skeptical that a model that is so reduced relative to the biology tells us much about underlying mechanisms in this case, particularly given it's reliance on a bespoke motif that is not well motivated by biological observations and completely ignores how other GABAergic populations could effect the dynamics of PV and PC firing. If the overall impact of the paper hinged on this I would not recommend publication without significantly more work extending the model or validating it in vivo. As it is, the experimental data speaks for itself and the model feels like a relatively inconsequential add-on suggesting one possible mechanism without convincingly demonstrating it. I don't want this to sound overly harsh, it is a a clever idea and I would be interested to read a computational work by the same group investigating how this motif functions within a more realistic dynamic environment, or that explains how the required asymmetric strong/weak competition circuits form in the first place, or makes testable predictions subsequently validated experimentally. In its current form, however, the model does little except say, essentially, "if PC cell firing is dominated by a small subset of GABAergic cells that are wired together in a very particular way, that could explain one aspect of our experimental observations".

Reviewer #2: The authors generally addressed my comments. I only have two remaining points:

1) In their rebuttal the authors provide a supplementary figure to clarify what they displayed in Fig. 5F-G. I think that this figure more transparent and clear in presenting the data and the small (yet significant) size of the effect the authors report. I strongly recommend to replace Fig. 5F-G with the panels that the authors included in their rebuttal. The statistics (values and types) should be included in the legend of the updated Fig. 5F-G.

2) The updated computational model is a significant upgrade. As a minor point, I suggest that the authors harmonize the y-axes in Fig. 6B-F, so that it is clearly visible what the different between the various conditions are. I was also rather surprised by the fact that the updated model yielded results that were basically undistinguishable from the earlier model (error bars may be slightly different, while the average values appear unchanged). I wonder if the authors have an explanation for this, or if it is just an aesthetic effect.

Reviewer #3: Based on the authors' responses to my questions as well as those posed by other reviewers, all of my concerns have been adequately addressed. Thank you.

---

## [Editor Report · Decision Letter 3]

7 Nov 2025

Dear Dr Poort,

Thank you for the submission of your revised Research Article "Context-dependent activation of V1 parvalbumin interneurons enhances visual discrimination" for publication in PLOS Biology. On behalf of my colleagues and the Academic Editor, Alberto Bacci, I am pleased to say that we can in principle accept your manuscript for publication, provided you address any remaining formatting and reporting issues. These will be detailed in an email you should receive within 2-3 business days from our colleagues in the journal operations team; no action is required from you until then. Please note that we will not be able to formally accept your manuscript and schedule it for publication until you have completed any requested changes.

PRESS

Sincerely, 

Taylor

Taylor Hart, PhD,

Associate Editor

PLOS Biology

thart@plos.org